# Interplay of Kelvin-Helmholtz and superradiant instabilities of an array of quantized vortices in a two-dimensional Bose-Einstein condensate

**Luca Giacomelli⋆ and Iacopo Carusotto†**

INO-CNR BEC Center and Dipartimento di Fisica, Università di Trento,
via Sommarive 14, I-38050 Povo, Trento, Italy

⋆ luca.giacomelli-1@unitn.it , † iacopo.carusotto@unitn.it

Ancillary videos associated to the data presented are available in arXiv:2110.10588 [25].

## Abstract

We investigate the various physical mechanisms that underlie the dynamical instability of a quantized vortex array at the interface between two counter-propagating superflows in a two-dimensional Bose–Einstein condensate. Instabilities of markedly different nature are found to dominate in different flow velocity regimes. For moderate velocities where the two flows are subsonic, the vortex lattice displays a quantized version of the hydrodynamic Kelvin–Helmholtz instability (KHI), with the vortices rolling up and co-rotating. For supersonic flow velocities, the oscillation involved in the KHI can resonantly couple to acoustic excitations propagating away in the bulk fluid on both sides. This makes the KHI rate to be effectively suppressed and other mechanisms to dominate: For finite and relatively small systems along the transverse direction, the instability involves a repeated superradiant scattering of sound waves off the vortex lattice; for transversally unbound systems, a radiative instability dominates, leading to the simultaneous growth of a localized wave along the vortex lattice and of acoustic excitations propagating away in the bulk. Finally, for slow velocities, where the KHI rate is intrinsically slow, another instability associated to the rigid lateral displacement of the vortex lattice due to the vicinity of the system's boundary is found to dominate.



# 1 Introduction

Parallel flows in hydrodynamics are known to give rise to instabilities, with one of the most fundamental being the Kelvin–Helmholtz instability (KHI) occurring in the *shear layer* separating two parallel uniform flows [1, 2]. The effect of this instability is to amplify perturbations located in the transition region and create a characteristic vortex-like flow pattern that grows until the two flows mix up in a turbulent way.

In incompressible fluids, the KHI occurs both for a vortex sheet (or tangential discontinuity), in which the velocity is discontinuous between the two uniform parallel flows (see for example §29 in [3]), and in the case in which the velocity shows a smooth transition. In this latter case, the effect of the finite width of the shear layer is to quench the instability at smaller scales [1, 2].

Even more exciting features are found when compressible fluids are considered, in particular for relative velocities $\Delta v$ of the order of twice the speed of sound in the fluid. For a vortex sheet in two spatial dimensions the KHI is suppressed and the discontinuity becomes stable again for $\Delta v > 2\sqrt{2}c_s$ (see Problem 1 in §84 of [3]). In the case of a finite-size shear layer, the instability is present for all values of $\Delta v$, but displays different properties for $\Delta v > 2c_s$ [4–8]. This change in behaviour is due to the appearance of negative-energy modes for acoustic perturbations in supersonically flowing compressible fluids.

This same feature also underlies the amplified reflection (or over-reflection) of acoustic waves at an interface with $\Delta v > 2c_s$ (Problem 2 in §84 of [3]). Within the gravitational analogy framework [9], this phenomenon can be related to superradiant scattering from rotating black holes [10], whose analog has been recently observed in a water tank displaying a draining vortex flow configuration [11]. In parallel flows instead, amplified reflection is typically associated to dynamical instabilities that complicate the picture (see for example discussion in §11.5 of [12]).

Since the KHI is an inviscid phenomenon, determined by inertial effects and not by viscosity, it can be expected to also take place in the inviscid flow of superfluids. In this context, KHI was experimentally observed at the interface between the two superfluid phases of $^3$He [13–15]. Subsequently, dilute Bose–Einstein condensates (BECs) of ultracold atoms have been theoretically considered for the study of this phenomenon: in particular, KHI was shown to develop in phase-separated two-component BECs [16–18], while the KHI of a quantized vortex array in a single component condensate was explored in [19]. Given the irrotational nature of the velocity field of a single-component BEC, the only way to accommodate the ve-

locity difference is in fact by creating an array of quantized vortices along the shear layer. More recently, a similar velocity field in a BEC with a different density distribution has been experimentally investigated in [20]. Here, a condensate subject to a synthetic magnetic field (obtained with a rotation of the trap) was prepared in a single Landau gauge wavefunction, displaying an elongated shape. This configuration was shown to undergo a snaking instability that was connected with a KHI and that leads to a crystallization of the BEC in droplets.

In this article we investigate the interplay of superradiant phenomena and the KHI of an array of quantized vortices in a single component atomic BEC confined between two hard-wall potentials. While for $\Delta v < 2c_s$ we recover and further characterize the KHI studied in [19], a much richer physics is found for $\Delta v > 2c_s$ when the KHI mixes with propagating sound waves and gets effectively quenched. In finite-size configurations along the direction transverse to the velocity, the KHI is replaced by a slower instability: the perturbation starts to appear in the bulk of the two counter-propagating flows and only at later times it produces a significant distortion of the vortex array. This is a superradiant instability (SRI) that can be understood in terms of a repeated amplified scattering of sound waves in the two surrounding bulk regions, that are then bounced back towards the vortex array by the edges of the system. An analogous mechanism was studied in [21] in a configuration where the velocity jump was created by a static synthetic gauge field with no independent dynamics.

While this mechanism cannot take place in an unbound system, where no repeated amplifications can occur, surprisingly we find yet another kind of instability in which *surface modes* localized in the shear layer grow together with acoustic waves propagating away in the two flows. This *radiative instability* (RI) is again superradiant in nature since it relies on the fact that the localized mode is resonant with travelling waves and has an opposite-signed energy with respect to them. As such it is analogous to the ergoregion instabilities of multiply quantized vortices [22, 23], that are also given by the resonance of negative-energy localized *core modes* with positive-energy propagating waves.

While in the KHI and SRI regimes the long-distance hydrodynamic picture qualitatively captures the dominating physical effects, in the RI regime the short-distance quantized nature of the shear layer becomes important for the existence of localized excitations on resonance with travelling ones. This quantized nature is also important when small velocities are considered and the dominating instability turns out to be associated to a rigid lateral displacement of the vortices. This *drift instability* (DI) gets faster when the system size is decreased and is analogous to the motion of a vortex near the boundary of a condensate confined with a hard-wall potential [24]. The presence of this additional instability however does not prevent the development of the KHI that continues to dominate the long-time evolution of the system.

The structure of the paper is as follows. In Section 2 we introduce the system under study and show the results of GPE numerical calculations for regimes displaying different kinds of instability. In Section 3 we shine more light on the problem by computing the linearized Bloch-waves Bogoliubov spectra for different velocities. In Section 4 we describe in detail the different instability regimes: in Section 4.1 we describe the Kelvin–Helmholtz instability (KHI) regime, in Section 4.2 we discuss the superradiant instability (SRI) and the radiative instability (RI), and in Section 4.3 we address the drift instability (DI) happening at small relative velocities. Finally, in Section 5 we draw the conclusions. As a further support to our conclusions, in Appendix A we display the result of additional numerical calculations confirming the occurrence of superradiant scattering in this system.

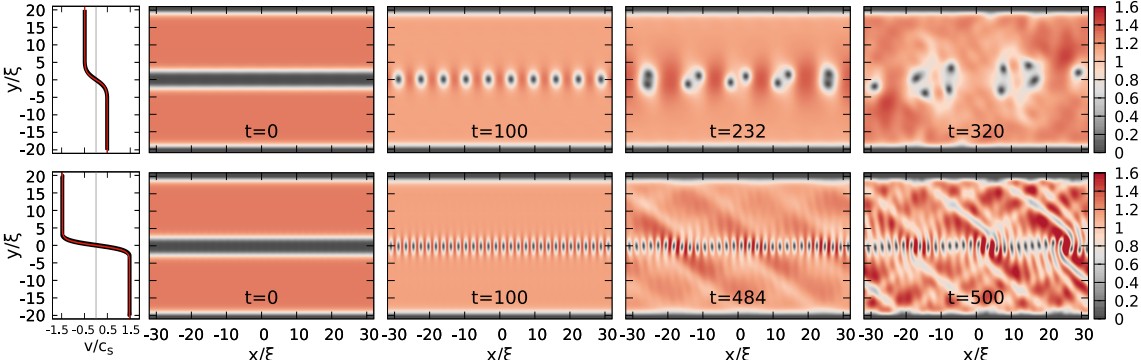

Figure 1: GPE simulations of the time evolutions of the density of a condensate confined along $y$ by hard walls at $y = \pm L_y/2$, with $L_y = 40\xi$ and with a central Gaussian potential $V_G = A\exp(-y^2/2\sigma^2)$ with $A = 5\mu$ and $\sigma = \xi$. Opposite velocities $\pm v = \pm\Delta v/2$ are imposed in the two regions $y < 0$ and $y > 0$ and, when the Gaussian potential is progressively lowered (between $t = 0$ and $t = 100\ \mu/\hbar$), an array of quantized vortices develops. The upper plots show the case $\Delta v = 0.98\ c_s$ and the lower ones $\Delta v = 2.94\ c_s$. In the leftmost panels we show cuts of the velocity profile along $y$ for a $x = 32\xi$ position located in between a pair of vortices for time $t = 100\ \mu/\hbar$ after the formation of the array of vortices: the thicker black lines show the numerical data, the thinner red lines are plots of equation (3), with the shear layer width (4). The upper panels show the KHI behaviour presented in [19], in which vortices cluster and co-rotate. The lower panels show an example of SRI, in which the instability is slower and the unstable mode is not localized on the vortex line, but is distributed all over the system. Times are expressed in units of $\hbar/\mu$. A video of these evolutions is available online [25].

## 2 GPE simulations

We consider an atomic BEC at $T = 0$ tightly confined in one direction so that the relevant dynamics takes place in two spatial dimensions. The condensate can be described with a two-dimensional GPE for the classical complex field $\Psi(t, x, y)$, describing the BEC order parameter [26]:

$$i\hbar\partial_t\Psi = \left[-\frac{\hbar^2\nabla^2}{2M} + g|\Psi|^2 + V_{\text{ext}}(t, y)\right]\Psi. \tag{1}$$

Here $M$ is the atomic mass, $g$ is the interatomic interaction constant and $V_{\text{ext}}$ is an external potential that, as indicated, we take to depend only on $y$ and to be possibly time-dependent. A stationary state of the GPE can be taken in the form $\Psi(t, x, y) = e^{-i\mu t/\hbar}\sqrt{n(x,y)}e^{i\Theta(x,y)}$, where $\mu$ is the chemical potential, $n$ is the density of condensed atoms and $\Theta$ is the phase, that is related to the velocity of the condensate by $\mathbf{v} = \hbar\nabla\Theta/M$. Interactions imply the presence of a finite sound speed, that in a constant-density condensate is $c_s := \sqrt{gn/M}$, and introduce a typical length scale, the so-called *healing length* $\xi := \hbar/(Mc_s)$.

As was done in [19], we consider periodic boundary conditions along $x$ and a potential composed by two hard walls at $y = \pm L_y/2$ and by a repulsive Gaussian potential centered in $y = 0$. This central Gaussian potential is initially strong enough so that the condensate is composed by two parts separated and independent, as shown in the $t = 0$ panels of Figure 1. The ground state of the GPE with this potential is computed via the conjugate gradient method described in [27] and at $t = 0$ the two condensates are given equal and opposite *momentum kicks* along $x$, so that they develop equal and opposite velocities $\pm v = \pm\Delta v/2$. The intensity of the central Gaussian potential is then linearly decreased in time so to vanish at $t = 100\ \mu/\hbar$,

after which the external potential is composed only by the external time-independent hard walls. Once the central barrier has been lowered, the only way for the condensate to satisfy the irrotationality of the velocity vector field is to create an array of (singly) quantized vortices along $y = 0$, in numbers equal to the difference of the winding numbers of the phase in the two regions. In mathematical terms, given a relative velocity $\Delta v$ between the two condensates, the number of vortices per unit length will be

$$n_{\text{vort}} = \frac{M}{\hbar} \frac{\Delta v}{2\pi}. \tag{2}$$

For a value of $x$ that corresponds to a half point between two vortices, the $y$ dependence of the velocity $v_x$ can in good approximation be fitted with a functional form

$$v_x(y) = \frac{\Delta v}{2} \tanh(y/\delta_{\Delta v}), \tag{3}$$

$\delta_{\Delta v}$ being the width of the transition region between the two counter-propagating uniform flows $\pm\Delta v/2$. This hyperbolic tangent velocity profile is well-known in hydrodynamics, whose KHI for the incompressible fluid case was studied in [28], and whose stability in the compressible fluid case was characterized in [4–6]. In the present case the shear layer width $\delta_{\Delta v}$ is not independent of the velocity difference, but behaves approximately as

$$\delta_{\Delta v} \simeq \xi \left(1 + \frac{c_s}{\Delta v}\right). \tag{4}$$

Two examples of the $y$ dependence of the velocity can be seen in the leftmost panels of Figure 1, where the numerical profile (black) is successfully compared to the functional form (3) shown in red.

The rest of Figure 1 shows, for two different values of the relative velocity, snapshots of the time-evolution of the GPE following this procedure; a full video showing these evolutions is available online [25]. In the first example, for $\Delta v < 2c_s$, the vortex line is unstable with a mechanism similar to the hydrodynamic KHI: after some time vortices start to move from the horizontal $y = 0$ line and begin to co-rotate in clusters of growing size, as first characterized in [19]. In the second example, for $\Delta v > 2c_s$, instead, even if the vortices are much closer, they do not initially move from the horizontal line and the unstable mode simultaneously develops in the whole system, as can be seen from the emerging density pattern. Surprisingly, the vortices take a longer time to move and this motion is associated to significant density variations in the *bulk* of the two flowing regions. The spatially oscillating shape of the emerging pattern indicates that the instability is due to a superposition of up-going and down-going propagating phononic waves.

Further evidence of the different localization of the unstable modes in the two cases is obtained by varying the transverse size of the system $L_y$, that is the separation between the two hard walls. We observe that for $\Delta v < 2c_s$ the time needed for the vortex line to deform is essentially independent from $L_y$, while for $\Delta v > 2c_s$ the instability rate decreases for increasing $L_y$. All these features suggest that the origin of the second kind of instability is the same of the superradiant instabilities (SRI) observed in [21], where a tangential discontinuity of the velocity was induced via a synthetic gauge field. These instabilities involve the repeated amplification of propagating modes of opposite energies and give rise to patterns such as the one visible in the $t = 484\mu/\hbar$ snapshot of the second row of Figure 1.

While showing that instabilities with different features occur, time evolutions of the GPE are not the best tool to obtain a complete picture of the underlying instability mechanisms. The long time that is required for the instability to significantly deform the line of vortices indicates that the configuration obtained by lowering the central Gaussian potential is a stationary state of the GPE. This is confirmed by the fact that imaginary time evolutions of the GPE with fixed

winding numbers in the two regions converge to states of the same shape as those found with the above time-dependent procedure.[1] In the following, we hence resort to a study of the linear stability of this stationary state with a Bogoliubov approach.

## 3   Bloch functions for the Bogoliubov problem

The natural approach to the study of the Bogoliubov problem in this configuration is to take advantage of the periodic structure of the stationary states we are interested in; the vortices along $y = 0$ are in fact equispaced along $x$, as can be seen in the second ($t = 100\,\mu/\hbar$) panels of Figure 1. We consider a small deviation

$$\Psi(t, x, y) = e^{-i\mu t/\hbar}\left(\Psi_{\Delta v}(x, y) + \delta\psi(t, x, y)\right),\tag{5}$$

where the stationary state $\Psi_{\Delta v}$, relative to a given velocity difference $\Delta v$, has the periodicity

$$\Psi_{\Delta v}\left(x + \frac{2}{n_{\text{vort}}}, y\right) = \Psi_{\Delta v}(x, y).\tag{6}$$

While the periodicity of the complex order parameter $\Psi_{\Delta v}$ is $2/n_{\text{vort}}$, what matters for the fluctuations are the density and the velocity, that instead have a period $1/n_{\text{vort}}$. This can also be seen from the fact that only the square of the order parameter enters in the linear problem (9) we are now going to write.

The perturbations (5) around a stationary state are described at the linear level by the Bogoliubov equations, that can be written, by considering $\delta\psi$ and $\delta\psi^*$ as independent variables [29], in terms of the Bogoliubov spinor[2]

$$\begin{pmatrix} \delta\psi(x, y) \\ \delta\psi^*(x, y) \end{pmatrix} = e^{iKx}\begin{pmatrix} \eta_K(x, y) \\ \chi_K(x, y) \end{pmatrix},\tag{7}$$

that we decompose in decoupled Bloch waves, where $\eta_K$ and $\chi_K$ are independent spinor components that have the periodicity $1/n_{\text{vort}}$ we just discussed, and $K$ belongs to the first Brillouin zone

$$-\frac{M}{\hbar}\frac{\Delta v}{2} \leq K \leq \frac{M}{\hbar}\frac{\Delta v}{2}.\tag{8}$$

The resulting Bogoliubov equations at given $\Delta v$ and $K$ are

$$i\hbar\partial_t\begin{pmatrix} \eta_K \\ \chi_K \end{pmatrix} = \begin{bmatrix} D_{\Delta v, K} & g\Psi_{\Delta v}^2 \\ -g\left(\Psi_{\Delta v}^*\right)^2 & -D_{\Delta v, K} \end{bmatrix}\begin{pmatrix} \eta_K \\ \chi_K \end{pmatrix} =: \mathcal{L}_{\Delta v, K}\begin{pmatrix} \eta_K \\ \chi_K \end{pmatrix},\tag{9}$$

with

$$D_{\Delta v, K} = -\frac{\hbar^2\nabla^2}{2M} - \frac{i\hbar K}{M}\partial_x + \frac{\hbar^2 K^2}{2M} + 2g|\Psi_{\Delta v}|^2 + V_{\text{ext}} - \mu.\tag{10}$$

The matrix involved in the Bogoliubov equations is not Hermitian, however it is $\sigma_3$-pseudo-hermitian, that is $\sigma_3\mathcal{L}_{\Delta v, K}^\dagger\sigma_3 = \mathcal{L}_{\Delta v, K}$, where $\sigma_3 = \text{diag}(1, -1)$. This implies that the evolution through the Bogoliubov equations conserves energy, but the energy of an eigenmode

---

[1]In the simulations of Figure 1 the departure from the stationary state is given by dynamical instabilities, that are seeded by the excitations that are generated by the time-dependence of the potential as well as by numerical noise.

[2]The choice of taking $\delta\psi$ and $\delta\psi^*$ as independent variables is a convenient one to obtain a linear problem for the fluctuations fields, whose governing equations are otherwise not linear since they mix $\delta\psi$ and its complex conjugate. This however doubles the number of degrees of freedom and, as a result, every eigenmode $(U_i, V_i)^T$ of frequency $\omega_i$ has a *particle-hole symmetric* mode $(V_i^*, U_i^*)^T$ of frequency $-\omega_i^*$. The physical fluctuation field $\delta\Psi$ is recovered by taking the sum of these two modes, so that $\delta\Psi = U_i e^{-i\omega_i t} + V_i^* e^{+i\omega_i t}$.

$|\phi_{K,i}\rangle = (U_{K,i}, V_{K,i})^T$ of $\mathcal{L}_{\Delta v, K}$ is not given simply by its frequency, but by

$$E_{K,i} = \left\| \phi_{K,i} \right\|_B \hbar \omega_{K,i}, \tag{11}$$

where $\left\| \phi_{K,i} \right\|_B := \int dx\, dy\, (|U_{K,i}|^2 - |V_{K,i}|^2)$ is the so-called Bogoliubov *norm* of the eigenmode. This can have both signs (and also be zero), so that for example negative-norm modes at positive frequencies have a negative energy; the presence of negative-energy modes is referred to as *energetic instability*. Zero-norm modes are instead associated to complex eigenvalues, that come in pairs of complex-conjugate frequencies; the modes with positive imaginary part of the frequencies are exponentially growing and are known as *dynamical instabilities*. These have zero energy and can emerge with the resonance of two opposite-normed modes, so that they can be thought as the simultaneous production of excitations with opposite energies.

To solve the Bogoliubov problem we first compute, for a given $\Delta v$, the order parameter with a conjugate gradient algorithm [27] on a numerical $x$ range of $2/n_{\text{vort}}$, imposing unit winding number in each region. We then construct the Bogoliubov matrix within half of the $x$ range and using discretized expressions for the derivatives. We diagonalize this matrix, for a given Bloch wavenumber $K$, and repeat the diagonalization to sample $K$ values throughout the first Brillouin zone.[3] Examples of the obtained spectra for different values of the velocity of the two opposite parallel flows are shown in Figure 2.

A general picture of these results can be obtained by neglecting for the moment the modes at the edge of the Brillouin zone, whose wavelengths is similar to the inter-vortex spacing. One can see that for $\Delta v < 2c_s$ the spectra are composed of positive-energy modes (positive-norm at positive frequencies and negative-norm at negative ones) and by a dynamically unstable branch with a vanishing real part of the frequency and an increasing instability rate when approaching $K = 0$. As we are going to discuss, these are the unstable modes responsible for the KHI. For $\Delta v > 2c_s$ instead, the positive- and negative-norm parts of the spectra merge, reflecting the energetic instability associated to the Landau instability of supersonic flows in both the upper and the lower regions of the system. As we have already anticipated, the pseudo-Hermitian nature of the Bogoliubov problem implies that when modes of opposite norm sign approach, they can give rise to a dynamically unstable branch.

The closing of the gap between the positive- and negative-norm modes for all Bloch wavenumbers perturbs the zero-frequency KHI branch and suppresses it. For high enough velocities, in fact, the KHI behaviour dominated by small Bloch wavenumbers is replaced by a lower dominating instability rate at finite $K$. The physical meaning of this behaviour of the spectra is that the modes responsible for the KHI can resonantly couple to collective modes of the two regions, so that the KHI is effectively *damped* by the emission of phonons in the two parallel flows. On the other hand, the modes at the edge of the Brillouin zone deviate from this general picture since they are the ones that are affected the most by the *quantized* nature of the shear layer. Many different effects can hence contribute to the physics, as we are now going to discuss in what follows.

A quick picture of the instability regimes can be obtained by looking at Figure 3 in which we show, for three different values of $L_y$, the maximum instability rate for different values of $v$ and the corresponding Bloch wavenumber. For intermediate velocities the maximum instability occurs for $K = 0$, increases linearly with the velocity and is independent on the system size; this is the KHI regime. For higher velocities $\Delta v > 2c_s$ the maximum instability rate occurs for finite values of $K$ (increasing with $\Delta v$), approaches a constant for increasing

---

[3]We checked that the spectra obtained in this way are robust with respect to variations of the numerical parameters, e.g. by changing the spatial discretization. The fact that the complex-frequency modes we find are not unphysical can also be checked by comparing the instability rates with the ones extracted from time evolutions of the Bogoliubov equations; an example of such a comparison in shown later on in the paper in the $L = 128\xi$ panel of Figure 5.

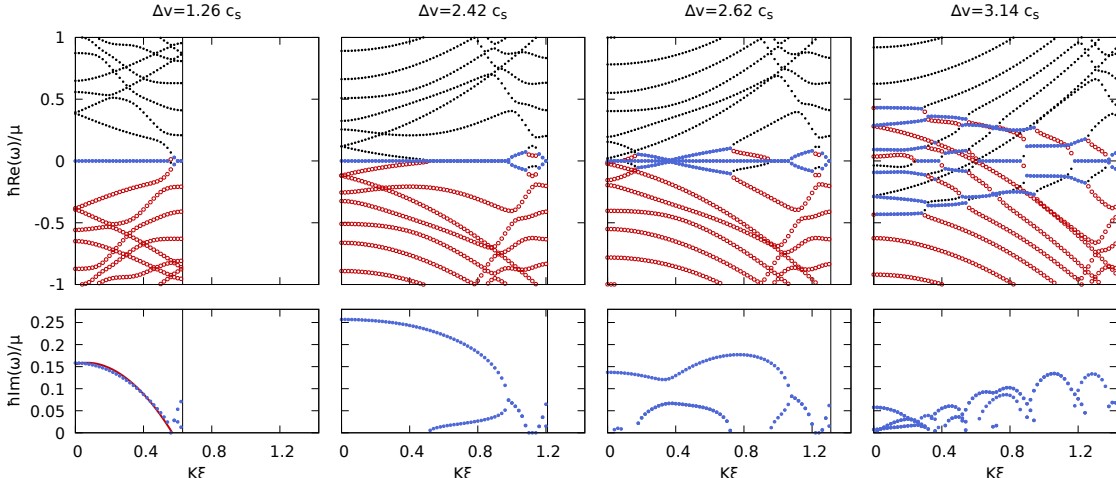

Figure 2: Real (top panels) and imaginary (bottom panels) parts of the eigenfrequencies of the Bogoliubov problem as a function of the Bloch wavenumber $K$ for transverse size $L_y = 20\xi$ and four different values of the relative velocity $\Delta v$. For each $\Delta v$ the $K$ range is truncated at the edge of the Brillouin zone. Black filled dots correspond to positive-norm modes, red empty dots to negative-norm ones and blue thicker dots to zero-norm dynamically unstable modes, giving rise to the *bubbles* of instability visible in the lower plots. When $\Delta v$ is high enough energetically unstable phononic modes begin to be present. Since $L_y$ is finite and hence the phononic spectrum discrete, this velocity threshold is larger that $2c_s$ and close to the $2.42c_s$ value used in the second panel. The spectra for values $1.26c_s < \Delta v < 2.42c_s$ are similar to the one for $\Delta v = 1.26c_s$. One can see the transition between the KHI regime, in which the dominating instability is at $K = 0$, to a regime of SRI, in which the instability maxima occur at finite $K$. For comparison, the red line in the leftmost lower panel is the hydrodynamic prediction (13), see Section 4.1 for details.

velocities and strongly depends on the system size; this is the SRI regime. The fact that the transition from KHI to SRI does not occur abruptly at $\Delta v = 2c_s$ is due to the finite transverse size of the system that results in discrete phononic modes in the two regions, so that higher $\Delta v$ are needed to have negative energy phononic waves. In the left panel of Figure 3 one can see that the width of the transition region is smaller for larger systems, so that one can expect the transition to occur exactly at $\Delta v = 2c_s$ in an infinite system, in which a continuum of phononic modes is available.

Besides these two regimes already observed from the GPE calculations, a third behaviour is visible at small velocities $\Delta v \lesssim 0.8c_s$, for which the maximum instability rate occurs for Bloch wavenumbers at the edge of the Brillouin zone and does not strongly depend on the relative velocity.

In the next Section we consider in detail each of these regimes.

# 4 Instability regimes

## 4.1 Moderate velocities: Kelvin–Helmholtz instability (KHI)

Before dealing with the new instability regimes we identified, it is worth to to take advantage of our Bogoliubov approach to further characterize the KHI mechanism first observed in [19].

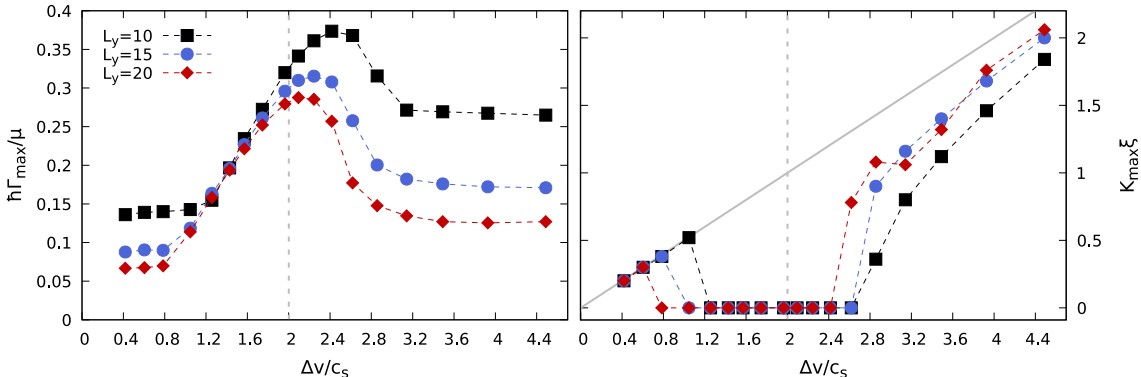

Figure 3: Instability rate (left plot) and corresponding Bloch wavenumber (right plot) of the most unstable mode (respectively $\Gamma_{max} = \max(Im(\omega))$ and $K_{max} = \text{argmax}_K(Im(\omega))$) as a function of the relative velocity $\Delta v$. Dashed lines are a guide for the eye, the solid gray line in the right plot indicates the edge of the Brillouin zone and the vertical dashed gray lines indicate $\Delta v = 2c_s$. The three symbols correspond to different transverse sizes of the system $L_y$, as indicated in the legend of the left plot.

To this purpose, we compare the numerical solution of the Bogoliubov problem with well-known analytical results of fluid mechanics, that emerge from a very different modeling the system.

In hydrodynamics the KHI for a (continuous) vortex sheet (i.e. a tangential discontinuity between two parallel flows $v_1$ and $v_2$) in an incompressible inviscid fluid of constant density and without gravity is known to have a dispersion relation of the form (see for example [1] Section 4.3.1, or [2] Section 4)

$$\omega(k_H) = \frac{v_1 + v_2}{2} k_H \pm i \frac{v_1 - v_2}{2} k_H, \tag{12}$$

where $k_H$ is the wavevector of the perturbation along the discontinuity, i.e. parallel to the flows. This result is modified if, instead of a tangential discontinuity, a finite-width shear layer is present; for example, for a piecewise continuous profile that is constant for $|y| > \delta$ and changes linearly for $-\delta \leq y \leq \delta$ the dispersion relation becomes (see for example [1] Section 4.3.2 or [2] Section 23)

$$\omega_{KH}(k_H) = \frac{v_1 + v_2}{2} k_H \pm i \frac{v_1 - v_2}{4\delta} \sqrt{e^{-4k_H\delta} - (2k_H\delta - 1)^2}. \tag{13}$$

At low wavenumbers $k_H$ the instability rate increases linearly, as in the zero-thickness case (12), while at higher transverse wavenumbers there is a decrease and above $k_H\delta \sim 0.6$ the instability is quenched. An analogous suppression of the instability was found in [28] for an hyperbolic tangent velocity profile, that, as already mentioned, well approximates the one of our superfluid case.

While these hydrodynamic models cannot be expected to describe the condensate we are considering, it is still interesting to compare qualitatively the features of equation (13) with our numerical results of Figure 2. First of all notice that, since our case corresponds to $v_1 = -v_2 = \Delta v/2$, the zero real part of the frequency of the KHI branch found in the numerics is in accordance with equation (13). Moreover, from the left plot of Figure 3, we see that between $\Delta v \sim 0.8\,c_s$ and $\Delta v \sim 1.6\,c_s$ the maximum instability rate increases linearly with $\Delta v$, in agreement with the linear dependence of the imaginary part of (13) on the relative velocity.

To compare the wavenumber dependence of the instability rate we need to take into account that our choice of the shape of fluctuations (5), in which the spatial phase of the stationary state is not taken as an overall factor, makes our Bloch wavenumber $K$ to differ from the hydrodynamic wavenumber $k_H$, that is instead measured with respect to the fluid. In particular, the hydrodynamic wavenumber $k_H$ is the one of the field $\phi$ in the expression $\Psi(t, x, y) = e^{-i\mu t/\hbar}\Psi_{\Delta v}(x, y)(1 + \phi(t, x, y))$. Also considering that the spectrum of (9) is even in $K$, Bloch wavenumbers in the portion of the first Brillouin zone we plot in Figure 2 correspond to hydrodynamic wavenumbers[4]

$$k_H = \frac{M}{\hbar}v - K\,.\tag{14}$$

Hence we should compare our results for the instability rates with the imaginary part of $\omega_{KH}(\frac{M}{\hbar}v - K)$, where $\omega_{KH}$ is given by (13). We plot this quantity with a red line in the leftmost lower plot in Figure 2, where we use $v_1 - v_2 = \Delta v$ and the width of the shear layer $\delta$ is taken as in equation (4). An additional shift in wavenumber is included to improve the similarity between the two results. Despite the need for this adjustment and the additional discrepancy in the unstable modes found at the edge of the Brillouin zone, the qualitative agreement with the magnitude of the instability rate and with the $K$-dependence of our numerical data is surprisingly good, considered that the hydrodynamic result is obtained from a very different model that can not be *a priori* expected to describe well our system. These similarities between the spectral features of the instability of our quantum superfluid and the ones of a standard KHI in classical hydrodynamics further justify the use of the KHI terminology in our context.

Notice that the suppression of the instability predicted in the hydrodynamic case at higher $k_H$ is instead not observed here. This can be interpreted with the fact that the threshold for the suppression $k_H \gtrsim 0.6/\delta$ is fixed by the width of the shear layer, that in our case is velocity-dependent and is given by (4); according to that scaling $0.6/\delta > \frac{M}{\hbar}\frac{\Delta v}{2}$, so that the hydrodynamic suppression threshold lies outside of the Brillouin zone. This means that the quantized nature of our shear layer removes the high-$k_H$ behaviour of the hydrodynamic prediction.

As a last comment it is interesting to note that, while the Bloch wavenumber $K$ of the fluctuation field $\delta\psi$ is the most natural parameter to consider in our approach, the spatial behaviour of density fluctuations $\delta n = 2Re(\Psi^*_{\Delta v}\delta\psi)$ is determined by $k_H$. Hence the $K = 0$ dominating instability in the KHI regime corresponds to a fluctuation $\delta\psi$ that (apart from the periodic part of the Bloch function) is constant throughout the system and thus corresponds to a density variation $\delta n = 2Re(\Psi^*_{\Delta v}\delta\psi) \sim \cos(\frac{M}{\hbar}\frac{\Delta v}{2}x)$, that correctly has opposite signs on neighboring vortices.

## 4.2 High velocities: superradiant (SRI) and radiative (RI) instabilities

### 4.2.1 Smaller systems: SRI

As we already discussed, the transition from the KHI regime to the SRI one by increasing the flow velocity can be seen in the three rightmost panels of Figure 2, in which the emergence of energetic instabilities associated to the supersonic flows above $\Delta v = 2c_s$ implies the existence of propagating phononic modes with which the $Re(\omega) = 0$ modes responsible for the KHI can couple, suppressing thus the instability.

The presence of negative-energy modes associated to supersonic motion signals the possibility of having superradiant scattering when these are resonant with a positive-energy wave: if a positive-energy wave impinges on the shear layer from one region and there is a resonant

---

[4]This relation depends solely on our choice for the analytical form of the fluctuation field, that we felt to be the most natural in our approach to the system. Use of this choice does not rely on any physical property of our superfluid system and an analogous choice could be made also in the classical hydrodynamics context.

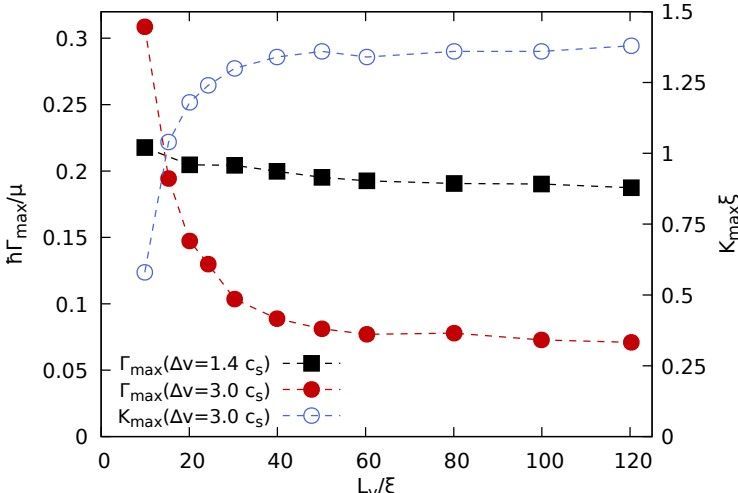

Figure 4: Dependence of the maximum instability rate $\Gamma_{max} = \max(Im(\omega))$ on the system size $L_y$ for $\Delta v = 1.4c_s$ in the KHI regime (black squares) and for $\Delta v = 3c_s$ in the SRI/RI regime (filled red circles); empty blue circles show the $K$-space location of the most unstable mode $K_{max} = \mathrm{argmax}_K(Im(\omega))$ in the second $\Delta v = 3c_s$ case ($y$ axis on the right).

negative-energy mode in the other region, the positive-energy wave can be reflected with increased amplitude. This is demonstrated in Appendix A.1, where we show how superradiant scattering can occur in the present setup for frequencies and Bloch wavenumbers $K$ for which energetic instabilities are present in the system. With respect to the simpler case of a tangential discontinuity [21], the presence of the array of vortices allows for more complex amplified scattering processes involving more waves.[5]

In a finite system along $y$, as in the case of Figure 1, amplified scattering processes will lead unavoidably to dynamical instabilities, since the reflection of the waves at the boundaries of the condensate will cause repeated amplification. This is the physical meaning of the dynamically unstable branches that occur when two eigenmodes of opposite norm (and hence opposite energy) approach the same frequency, as can be seen in the plots of the real part of the frequency of Figure 2: the two modes correspond to travelling waves in the two regions; whenever they are resonant, they can give rise to repeated amplified scattering. It is interesting to comment that this mechanism based on the mixing of modes of opposite energies is also responsible for superradiant scattering and superradiant instabilities in rotating spacetimes and their fluid analogues [10, 21].

The result of this linear instability is the behaviour shown in the rightmost panel in the second row of Figure 1, the emerging pattern being due to the superposition of the up-going and down-going waves of opposite energies, whose wavelength along $x$ is reflected in the maxima of the instability rate at finite values of $K$ visible for example in the rightmost plot of Figure 2.

Given this picture, one expects the instability rate to depend on the *round-trip time* of excitations in the two regions, that is to decrease when the transverse size of the system $L_y$ is

---

[5]Notice that the $\Delta v > 2c_s$ condition for the onset of the SRI has different meanings in different cases. In hydrodynamic shear layers, in which the shear layer has a full translational invariance along $x$, this threshold can be understood from the fact that for lower relative velocities a reference frame exists in which the flow is everywhere subsonic [21]. In the present discrete case instead the quantized vortices break the Galileian invariance along $x$, since there is a single reference frame in which they do not move; superradiant phenomena can hence only occur when supersonic flows with respect to this reference frame are available.

increased. To verify this, we repeated the computation of the Bogoliubov spectra as in Figure 2 for a given $\Delta v$ and different values of $L_y$. In Figure 4 we show the maximum instability rate for each size of the system. The black squares are the result for a velocity in the KHI regime, that, as expected, shows little dependence on the system size; the red filled circles are instead the result for a velocity in the SRI regime. One can see that in this case the instability rate decreases quickly with the system size, as expected. Still, quite unexpectedly, for even larger $L_y$ it approaches a finite value, and not zero as one would expect for a SRI.

### 4.2.2 Larger and infinite systems: RI

To get a better picture of what happens at larger sizes, in Figure 4 we also show a plot of the $K$ corresponding to the maximum instability rate (blue empty circles) and in the left panels of Figure 5 we show the instability rates of all dynamically unstable modes for four different values of $L_y$. One can see that, while $L_y$ is increased, the instability *bubbles* increase in number, corresponding to the increased density of modes in the larger system, that gives rise to more dynamically unstable branches. Moreover the height of these bubbles decreases, as one would expect for a SRI. However, for large systems, a branch of unstable modes becomes visible, that depends little on the system size, as can be seen by comparing the $L_y = 60\xi$ and the $L_y = 120\xi$ plots in Figure 5. From this comparison one can expect instabilities in the infinite system at all $K$, with the maximum instability rate in the rightmost bubble, near the edge of the Brillouin zone. This branch of unstable modes are what we call RI.

The behaviour of the spectrum in an infinite system can be inferred by comparison with the result of the diagonalization of the largest system we considered, $L_y = 120\xi$. In the right panel of Figure 5, all the unstable modes obtained with the diagonalization are indicated as dots, their color expressing their instability rate. The red shaded area is the region in which one expects energetic instabilities in the two infinite uniform regions.[6] One can see that the majority of the unstable modes fall in this region; these are the least unstable modes and correspond to the many points visible under the main unstable branch in the $L = 120\xi$ panel in the left part of Figure 5. This is the portion of the spectrum responsible for the SRIs, whose instability rate decreases with the system size and vanishes in an infinite system, leaving only energetic instabilities that allow for stable superradiant scattering, as demonstrated in the appendix A.1.

The most unstable modes fall instead on the blue line. These are modes that are expected to remain dynamically unstable also in the infinite system. The part of the spectrum at the edge of the Brillouin zone in which the branches at finite $Re(\omega)$ join at $Re(\omega) = 0$ corresponds to the highest bubble visible in the imaginary part of the spectrum. Notice that this RI branch exits from the *superradiant* red region; this means that positive-energy phononic propagating modes are involved.

The fact that this branch remains dynamically unstable in the infinite system can be confirmed by a time-dependent simulation of the fluctuations on top of the flowing configuration, that evolve according to the two-dimensional Bogoliubov equations (9). Instead of working at fixed Bloch wavenumber, as we did for the diagonalizations, we take $K = 0$ in (9) and we sample many Bloch wavenumber values by considering a background composed by many *lattice cells*. We also start the evolution from a noisy configuration, so to seed instabilities. Absorbing regions for the fluctuations are included near $\pm L_y/2$ to simulate an open system in the $y$ direction and avoid SRIs. Snapshots of the resulting time evolution of the density variations $\delta n = 2Re(\Psi_v^*(U + V^*))$ is shown in Figure 6; a full video is available online [25].

One can see that the system is dynamically unstable from the fact that a particular density

---

[6]The two lines delimiting the region are given by $\pm\omega(k_x = \frac{M}{\hbar}\frac{\Delta v}{2} - K, k_y = 0)$, where the frequency is given by the Bogoliubov dispersion relation (15) of sound excitations in a uniform condensate.

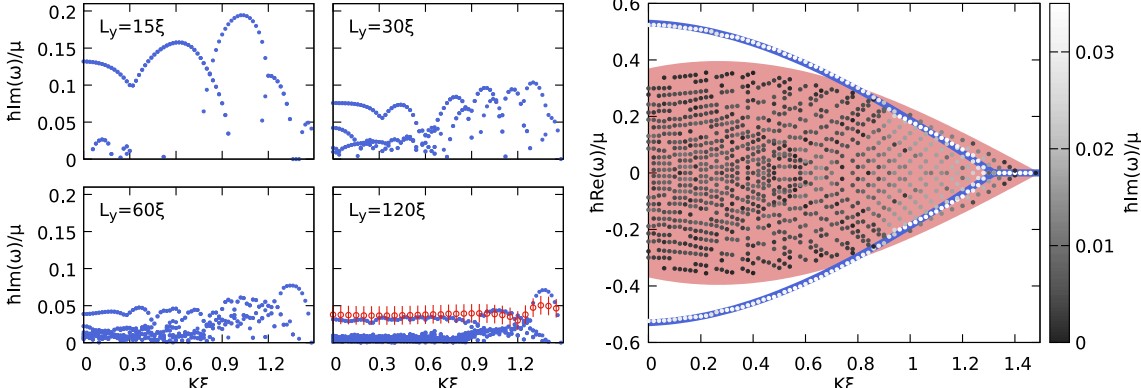

Figure 5: Left plots: instability rates of all the dynamically unstable modes for $\Delta v = 3c_s$ and four different values of the transverse system size. For $L_y = 120\xi$ we also show with red empty circles the instability rates extracted from the time-dependent calculations of Figure 6. The error bars are given by the finite evolution time of our simulations. Right plot: schematics of the spectrum of the infinite system, inferred from the distribution of the dynamically unstable modes of the $L_y = 120\xi$ system, that are plotted with points. The colorscale of the points shows their instability rate. The red filled area is where energetic instabilities associated to supersonic motion are expected in an infinite system. The blue line highlights the most unstable modes, responsible for the RI, that also remain unstable in the infinite system. In the surrounding white space only positive-energy modes are available.

perturbation is selected from the initial noise and quickly grows in time. The exponential form of the growth of the different wavenumber components is visible in the last panel of Figure 6. The density perturbation has a component that is spatially peaked around the vortices, composed by waves travelling along the quantized shear layer. Together with these localized excitations, also waves propagating away from the shear layer are present, visible in the striped pattern in the bulk of the flowing regions. Note that this pattern differs from the interference pattern visible in the second row of Figure 1, since waves propagating towards the shear layer are not present here. This time-dependent simulation hence confirms what was expected from the analysis of the spectra, namely that the instability corresponds to a simultaneous creation of negative-energy surface waves and of positive-energy sound waves that radiate away, hence the name RI.

While different from the SRI, this instability is still superradiant in nature, since it is based on the repeated amplification of the modes localized in the vortex array, at the expense of waves of opposite energy that are emitted away. In contrast to the SRI this instability does not need a boundary to trap part of the amplified excitations, that are automatically bound in the center of the system, and therefore is also present in an infinite system. Interestingly, this RI is similar to the *ergoregion instability* of multiply quantized vortices [23], in which the splitting into singly quantized vortices is driven by the simultaneous growth of negative-energy modes localized in the central core and a positive-energy one propagating away. In spite of the dynamical instability, superradiant scattering of waves can still occur along the lines of [23]. Differently from the SRI case, however, the amplified scattering is immediately followed by the quick growth of the RI. Further details on this process are given in Appendix A.2.

A quantitative comparison with the results of the diagonalization can be obtained by extracting the instability rates from the numerical time evolution. By means of a spatial Fourier transform, we extract the time evolution of the different $k_x$ components of the fluctuations. For each $k_x$ in the first Brillouin zone, we then fit the time dependence of the largest peak with

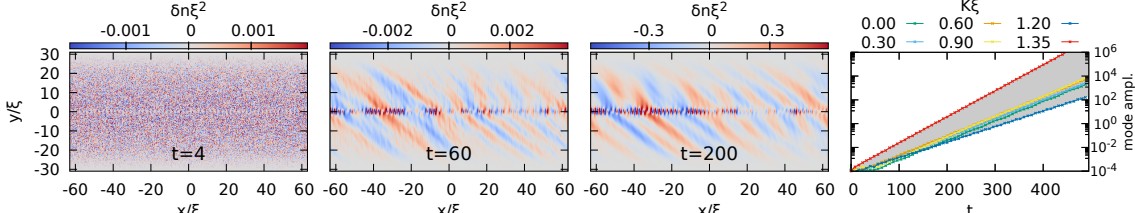

Figure 6: First three panels: snapshots of the time evolution of the density variations obtained by evolving in time the Bogoliubov equations for $\Delta v = 3c_s$ starting from a noisy configuration. Two absorbing regions with a Gaussian spatial profile around $\pm L_y/2$ are included to simulate an infinite system. The growing unstable modes is composed by waves travelling along the array of vortices and outgoing sound waves. Times are expressed in units of $\hbar/\mu$. A video of this evolution is available online [25]. Last panel: log-plot of the amplitude of some wavenumber components of the growing perturbations; one can see the exponential growth. The gray area indicates the interval between the lowest and the highest fitted rates.

an exponential curve. Some of these fits are shown in the last panel of Figure 6. The resulting growth rates are shown as red empty circles in the $L_y = 120\xi$ panel of Figure 5, where the error bars display the uncertainty given by the finite length of $T = 500\mu/\hbar$ of our simulation. One can see that these values compare well with the largest instability rates obtained for a finite but large system, apart from a deviation in the rightmost bubble that contains the dominating instabilities. This difference can be ascribed to a finite size effect in the diagonalization result; the height of that bubble can in fact be seen to be higher for the smaller $L = 60\xi$ system, and can hence be expected to further decrease in the limit $L_y \to \infty$. Apart from these minor quantitative deviations, this comparison confirms the expectation that the dynamically unstable branch in the right panel of Figure 5 is still present in an infinite system, and can hence be associated to the RI.

Notice that, while the SRI does not rely on the quantized nature of the shear layer and can be expected to take place in generic compressible fluids, the RI crucially relies on the small scale structure for the existence of the interfacial waves. Conceptually similar mechanisms have been predicted for incompressible fluids with density stratification, in which internal waves can have negative energy and interfacial gravity waves can exist. Configurations displaying SRI and others displaying RI have been considered (see §4 in [12] for a discussion). Interestingly, the system under study in the present work naturally displays both instability mechanisms in the context of compressible fluids.

## 4.3 Small velocities: drift instability (DI)

Up to now we characterized the regimes of instability occurring in the vicinity of $\Delta v = 2c_s$. However, while further decreasing the velocity, a marked deviation from the linear KHI decrease of the instability rate occurs for small velocities $\Delta v \lesssim 0.8c_s$, as can be seen in the left panel of Figure 3. The modes responsible for this change of behaviour are already visible in the leftmost panel of Figure 2, where a deviation from the main KHI branch is visible at the edge of the Brillouin zone $K = \frac{M}{\hbar}\frac{\Delta v}{2}$. We call the instability given by these modes *drift instability* (DI), since we are going to see that its main effect is to make the vortex array to drift laterally.

The instability rate of these modes is essentially independent on the relative velocity $\Delta v$, as can be seen in the upper panels of Figure 7, where the imaginary part of the frequencies of the unstable mode are shown for even smaller velocities than the ones considered in Figure 2. DIs hence do not depend on the spacing between the vortices. Differently, the KHI maximum

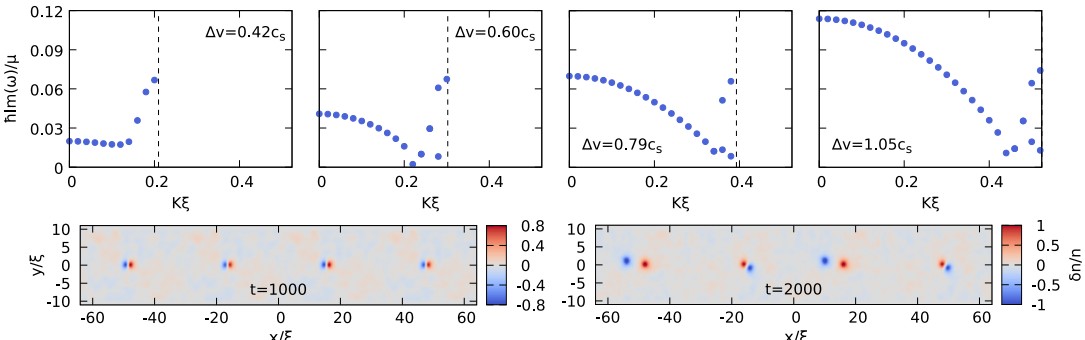

Figure 7: Upper panels: Plots of the instability rates analogous to Figure 2, for $L_y = 20\xi$ and smaller relative velocities $\Delta v$. While lowering $\Delta v$, the system goes from a KHI regime (last panel $\Delta v = 1.05c_s$) to a regime in which the instability is dominated by the modes near the edge of the Brillouin zone and is essentially independent of the velocity. Lower panels: Snapshots of the late time evolution of the density variation obtained by solving the GPE for $\Delta v = 0.2\,c_s$ and $L_y = 20\,\xi$. The first snapshot at $t = 1000\,\mu/\hbar$ shows the DI dominating at small velocities at early times, while the second one at $t = 2000\,\mu/\hbar$ shows the subsequent development of a KHI.

at $K = 0$ diminishes with the velocity, until its instability rate falls below the one of the modes on the edge of the Brillouin zone. From Figure 3 one can also see that, differently from the KHI that is essentially size-independent, the dominating instability at small velocities is stronger for smaller systems.

To get a better physical picture of DI we performed time evolutions of the GPE, analogous to the ones of Figure 1, for smaller relative velocities and longer times. While the long time fate of these evolutions is always of the KHI kind, with the vortices that move away from the central line and begin to corotate, at earlier times the behaviour is different.

This can be seen in the lower panels of Figure 7, where we show two snapshots of the density difference with respect to the initial state just after the complete formation of the quantized shear layer. Initially, density fluctuations grow in the same way around each vortex and correspond to a rigid drift of all vortices along the central line, hence the name DI. This is the effect of the fluctuations in the unstable modes on the edge of the Brillouin zone $\delta\psi \propto e^{i\frac{M}{\hbar}\frac{\Delta v}{2}x}$, that in fact correspond to density fluctuations $\delta n = 2Re(\Psi^*_{\Delta v}\delta\psi) \sim \cos(2\frac{M}{\hbar}\frac{\Delta v}{2}x)$, with a periodicity equal to the distance between the vortices.

The origin of this drift can be understood by performing additional numerical evolutions of the GPE with the vortex array placed at a different vertical position. What we observed (not shown) is that the direction of the horizontal motion depends on the location of the nearest boundary. In the present case in which we consider a velocity directed to the left (right) in the upper (lower) region, we found that the vortices move to the right if placed nearer to the upper boundary and to the left if placed near the lower boundary.

This indicates that the rigid drift of the array has the same physical origin as the motion of a single vortex near a hard-wall boundary of the condensate, where the wavefunction of the condensate vanishes. This was studied in [24], where a semi-infinite uniform condensate was considered. It was found that a vortex near the boundary moves parallel to it and that this motion can be understood by considering an *image vortex* with opposite circulation on the other side of the boundary, so that the motion of the vortex is analogous to a vortex-antivortex pair moving in parallel. While only this mechanism is active in our configuration, the precession of vortices would be a more complex effect in smoother, e.g. harmonic, traps, where it is also driven by the smooth density gradient towards the edge of the cloud [30, 31].

For a hard-wall instead the density reaches its bulk value within one healing length from the boundary and the effect of the density gradient is (exponentially) smaller.

In the present case we have two hard-wall boundaries, at equal distances from the vortices, so that the array can equally well drift in either direction, along either one of the image arrays of antivortices on the sides. One can hence think that the instability at small velocities stems from vertical fluctuations of the vortex array, that make it approach one of the two boundaries. The direction along which the array drifts will hence be given by the initial conditions in the fluctuations.

While being faster than the KHI for these velocities, this drift of the vortices does not prevent KHI from developing. This can be seen in the second snapshot in the lower part of Figure 7, where the density fluctuations are clearly seen to change periodicity. The *hydrodynamic* instability of the shear layer hence continues to dominate also at small velocities at late times, with the added physics of the lateral drift of vortices, that is obviously absent in the continuous hydrodynamic models.

## 5 Conclusions

In this paper we investigated the stability properties of the quantized shear layer occurring at the interface between two counter-propagating flows in an atomic two-dimensional BEC and found a rich interplay between different instability mechanisms as a function of the relative velocities $\Delta v$.

At moderate relative velocities $\Delta v < 2c_s$ we found, as already predicted in [19], an instability analogous to the hydrodynamic Kelvin–Helmholtz instability (KHI) with the vortices displacing from their initial position and corotating. At smaller relative velocities, a drift instability (DI) involving a lateral rigid displacement of the vortex array, analogous to the motion of vortices near a sharp boundary of the condensate, occurs on time scales shorter than those of the KHI, that however continues to dominate the late time fate of the shear layer.

At higher relative velocities $\Delta v > 2c_s$ instead, the KHI behaviour stops occurring due to the opening of phononic channels in which the modes responsible for the KHI can decay and different instabilities emerge. In a small enough system along $y$, the instability develops with excitations accumulating on each side of the shear layer. This instability has its origin in the existence of negative energy modes in the system, associated to supersonic motion of the condensate, that can give rise to superradiant scattering of sound excitations. The repetition of this process gives rise to superradiant instabilities (SRI). If an infinite system along $y$ is considered, the SRI is replaced by a radiative instability (RI) in which there is a simultaneous growth of interface waves travelling along the vortex array and of sound waves radiating away from it.

The mechanism responsible for SRI can be seen, through the ideas of analogue gravity, to be the same of superradiant instabilities in rotating spacetimes [10]. The occurrence of this kind of instability in the present system could be easily predicted from the point of view of our earlier work [21], in which a similar configuration (but without vortices) was considered to gain insight into the physics of superradiant scattering. The present work is hence an example in which analogies can help to look at problems from different perspectives.

From this same point of view, the mechanism of the RI instability, involving propagating waves and localized excitations, can be thought as something that could happen in a rotating spacetime in which the ergosurface has some extra structure that can host localized modes. The interplay with the SRI also suggests that the presence of this extra structure does not prevent the usual superradiant phenomena to happen. We are going to pursue these ideas in future work.

While the suppression of the KHI and the SRI can be expected to occur in any compressible inviscid fluid, since they essentially rely on the properties of sound waves, the existence of the RI depends on the specific structure of the shear layer and on its ability to support surface waves. It is also interesting to notice that the reliance of the RI on the resonance between a localized excitation mode with propagating sound modes makes it similar in nature to the instabilities of multiply quantized vortices [23].

Even if in the present work we focused on atomic Bose–Einstein condensates, this physics can be expected to also occur with similar phenomenology in other systems such as quantum fluids of light [32], whose superfluid properties are under active study and whose controllability could also allow experimental investigations of the instabilities that we characterized here.

To conclude, we showed that the flow around an array of quantized vortices displays an intriguing interplay between instabilities of different nature, with rich connections to various phenomena in different contexts, from well known behaviours of quantized vortices in trapped condensates, to classic hydrodynamic instabilities, to the physics of quantum fields in curved spacetimes.

# Acknowledgements

We acknowledge financial support from the H2020-FETFLAG-2018-2020 project "PhoQuS" (n.820392) and from the Provincia Autonoma di Trento. LG thanks Giulia Piccitto for computational help.

# A  Superradiant scattering

In this Appendix we show the occurrence of amplified scattering for suitably prepared wavepackets when the relative velocity is above the $\Delta v > 2c_s$ threshold. This gives further evidence of the superradiant nature of the instabilities we analyzed. Two kinds of superradiant scattering are possible here: the amplification of phononic waves at the expense of phononic waves of the opposite energy and the amplification of positive-energy phononic waves at the expense of the negative-energy modes localized around the vortex array. The first kind of amplification is responsible for the SRI, while the second one for the RI.

## A.1  Superradiant scattering responsible for the SRI

To understand the different kinds of scattering that can occur at the vortex array one can consider a large enough system, so that, far enough from the vortices, small amplitude fluctuations have a conserved momentum and obey the Bogoliubov dispersion relation that can be derived for a uniform condensate

$$\hbar\omega(k_x, k_y) = \hbar v_x k_x \pm \sqrt{\frac{\hbar^2 \mathbf{k}^2}{2M}\left(\frac{\hbar^2 \mathbf{k}^2}{2M} + 2gn\right)}. \tag{15}$$

The first term is a Doppler shift due to the velocity of the condensate, that is $v_x = +\Delta v/2$ for $y < 0$ and $v_x = -\Delta v/2$ for $y > 0$.

We are interested in scattering events in which a wave approaches the array of vortices from one of the two regions. In these scattering events the frequency is conserved, while $k_x$ is not. What is conserved is instead the Bloch wavenumber $K$ in the first Brillouin zone; when looking for the waves that are involved in the scattering one should hence also consider

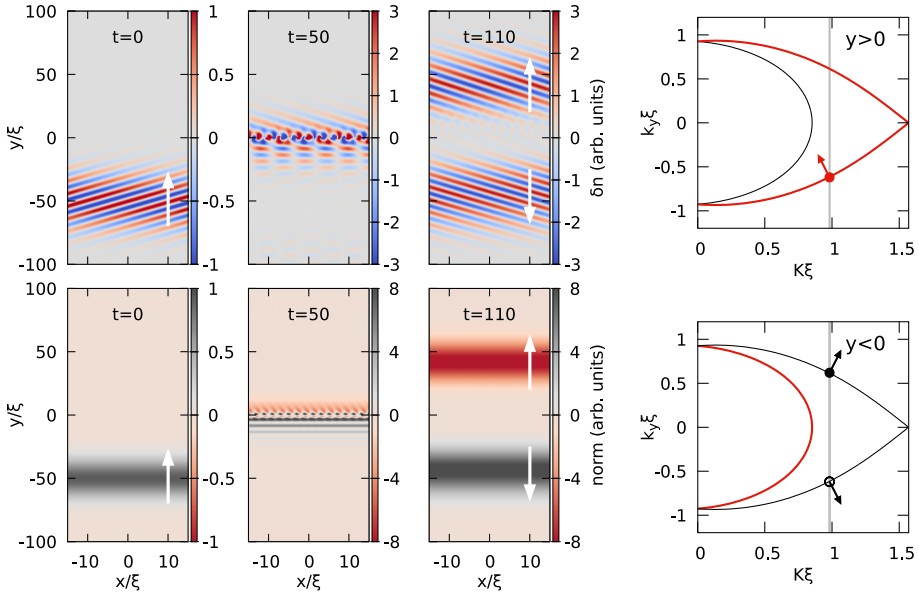

Figure 8: Left panels: snapshots of the time evolution of the Bogoliubov equations starting from an initial wavepacket at $\omega = 0$ with a wavevector chosen to travel towards the array of vortices. In the top row we plot the density variation $\delta n$ with respect to the stationary state $\Psi_{\Delta v}$, in the bottom one we plot the pointwise Bogoliubov norm $|U(x, y)|^2 - |V(x, y)|^2$ of the excitations. White arrows indicate the direction of motion of the wavepackets. Right panels: dispersion relation in the first Brillouin zone at fixed $\omega = 0$ in the uniform regions far enough above and below the array of vortices. Black thinner lines are positive-norm modes, red thicker lines negative-norm ones. The gray vertical line indicates the conserved Bloch wavenumber at which we are working, and the dots indicate the modes involved in the scattering process, with the arrows pointing in the direction of their group velocity. The considered case corresponds to $\Delta v = 3.14 c_s$ and gives rise to the kind of superradiant scattering of phononic waves that is at the basis of the SRI.

the extra modes that may be available at fixed $K$ due to the periodicity of the problem. In practical terms, besides $\omega(k_x, k_y)$ given by equation (15), one also needs to consider all other $\omega(k_x + n\Delta v, k_y)$, with $n$ integer.

In the right panels of Figure 8 we show cuts at $\omega = 0$ of the dispersion relation. The lower panel refers to the uniform region below the vortices, the upper panel to the one above them. Black thin lines are positive-norm modes and red thicker lines are negative-norm ones, that respectively correspond to the plus and minus signs in equation (15). Notice that the horizontal axis is restricted to the first Brillouin zone and the presence in each region of modes of both norms for some values of the Bloch wavenumbers is due to the periodicity of the system along $x$.

Suppose we consider the scattering of a positive-norm packet in the lower region peaked on the filled black dot on the dispersion curve. The other two dots show the available modes at the same frequency $\omega = 0$ and Bloch wavenumber $K$ that will take part in the scattering (with the arrows indicating the direction of their group velocities $\mathbf{v}_g = \nabla_{\mathbf{k}}\omega$): the initial packet will be transmitted to the negative-norm outgoing mode indicated with a red dot and will be reflected on the positive-norm mode indicated with a black circle. Because of norm (i.e. energy) conservation, the reflected packet will have a larger amplitude than the ingoing one.

This superradiant scattering can be verified with a time evolution of the Bogoliubov equa-

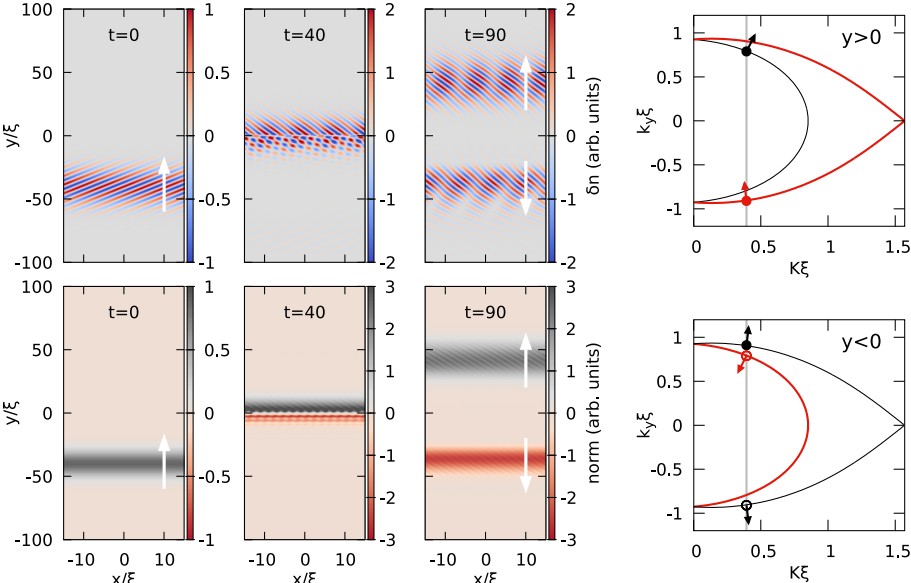

Figure 9: Same calculation as in Figure 8 for the same $\Delta v = 3.14 c_s$ and $\omega = 0$, but a different value of the conserved Bloch wavenumber $K$. For this choice, additional modes enter into play. The result is a superradiant process in which the incident positive-norm packet is amplified in transmission, while a negative-norm packet is reflected. The presence of the additional modes is signalled by the weak oblique fringes that are visible on top of the wavepackets in the lower panel for $t = 90 \hbar / \mu$.

tions (9), taking as an initial condition a wavepacket in the lower region that is a plane wave along $x$ and has a Gaussian profile along $y$, centered in wavenumber around the black dot on the corresponding dispersion plot and wide enough in space so that $\omega = 0$ is the most relevant frequency and the overlap with the dynamically unstable RI modes is negligible. Snapshots of this time evolution are shown in the left part of Figure 8, where the corresponding variations of the condensate density are shown in the upper row, while the pointwise norm of the fluctuations $|U(x,y)|^2 - |V(x,y)|^2$ is shown in the lower row. One can see that the scattering occurs as expected: the initial packet has a positive norm, is transmitted to a negative-norm packet and is correspondingly reflected in an amplified way (notice the different colorscales of the second and third snapshots).

This kind of scattering is completely analogous to the one we investigated in [21] in a configuration where the motion of the condensate is given by an externally applied synthetic gauge field, that allows to have a shear layer without quantized vortices. However, the periodic structure due to the presence of vortices in the present setup also gives rise to different scattering events. As an example, in Figure 9 we show the scattering of a wavepacket of the same frequency $\omega = 0$ but with a different $K$, at which also another branch of modes is available in both regions due to Bloch's theorem. There are now two transmitted modes and two reflected modes, of both norm signs. Correspondingly, differently from the previous case, one can see in the snapshots for $t = 90$ that the packets show fringes due to the simultaneous presence of more than one mode. But the most striking difference is that the main process in this case is the reflection of a negative-norm packet and the transmission of an amplified positive-norm one (again notice the different colorscales), so superradiance is now happening in transmission instead of reflection.

Despite this curious physics added by the discrete nature of the shear layer, these simulations clearly show the occurrence of sizable amplification of the wavepackets. A finite size of

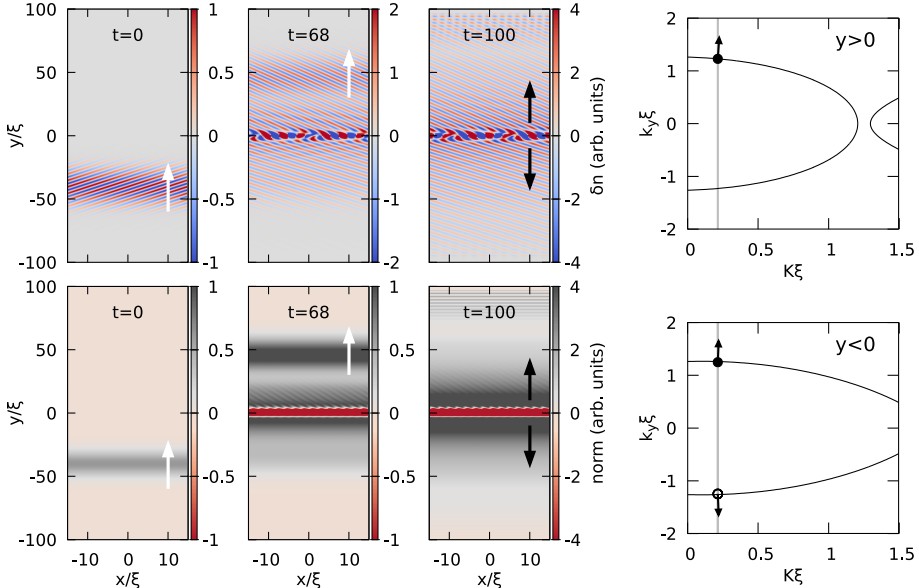

Figure 10: Same calculation as in Figures 8 and 9 for different values of $\Delta v = 3.14 c_s$ and $\omega = 0.5\mu/\hbar$. Here only positive-energy modes are available in the two asymptotic regions, so that scattering results in an amplified transmission together with the seeding of the RI. The white arrows indicate the direction of motion of the incident and transmitted wavepackets, while the black arrows indicate the propagation direction of the phononic waves due to the RI.

the system will cause these amplified packets to be reflected back to the shear layer, where they undergo further amplified scattering. The repetition of this process gives rise to superradiant instabilities, that occur for frequencies and Bloch wavenumber at which phononic waves of both norms coexist in the two regions, that is in the red region of the right panel of Figure 5. These growth rates of these dynamical instabilities decrease while increasing the system size, vanishing in an infinite system, where they only give rise to the dynamically stable amplified scattering that was demonstrated in this Appendix.

## A.2 Superradiant scattering responsible for the RI

Let us now consider a different frequency, at which no negative-energy phononic waves are available, but at which the dynamically unstable branch responsible for the RI is present, for instance one of the points of the blue line of the right panel of Figure 5 that do not fall in the red region. An example of the dispersion relations at fixed frequencies in the two regions is shown in the right panels of Figure 10.

We take as initial condition a packet that is a plane wave along $x$, with wavenumber indicated by the gray line, and peaked in $k_y$ at the filled black dot on the $y < 0$ dispersion curve. The evolution of the density fluctuations and of the corresponding pointwise norm are shown the snapshots on the left. As expected, the initial positive-energy wavepacket is transmitted as a positive-energy wavepacket in the upper region, as indicated by the white arrow in the second snapshots for $t = 68$. The amplitude of this packet is visibly larger than the incident one.

This amplification is due to the presence of a resonant negative-energy mode localized near the array of vortices, that is clearly visible in the second and third snapshots of the pointwise norm. The subsequent dynamics is however very different from the previous cases. Since the localized mode is resonant with phononic modes modes of opposite energy sign in both

regions, it will continue to grow in time while emitting these waves (black arrows in the last snapshots): this is the RI.

Differently from superradiant scattering of phononic waves, this superradiant process does not need a finite system to become unstable since the negative-energy modes are *trapped* and thus automatically subject to repeated amplification. This behaviour is analogous to the one of multiply quantized vortices, whose splitting instability is driven by the superradiant amplification of a negative-energy mode localized in the core of the vortex when this is resonant with some phononic wave in the rest of the condensate [23].

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
