# Peer review of "Interplay of Kelvin-Helmholtz and superradiant instabilities of an array of quantized vortices in a two-dimensional Bose-Einstein condensate"

_SciPost Physics, doi:SciPost Phys. 14, 025 (2023)_

## Round 2 · Referee Report · Anonymous (Referee 1) · 2022-1-17

Report
The authors present a numerical and theoretical study of the instabilities occurring at the interface between two superfluids, examining a range of relative velocities from subsonic to supersonic. They find a range of novel behaviours that they investigate both through GPE simulations and diagonalisation in the Bogoliubov approximation.
The paper extends the numerical results of Ref. [12], which used GPE simulations to investigate a similar geometry. Through the more complete analysis presented here, considerable additional insight is gained into the different regimes, while matching the qualitative results of Ref. [12] in the subsonic regime. In this regard, the paper is an important contribution to the study of shear-layer instabilities in 2D superfluids and will certainly serve as an important reference for future experimental realisations.
Despite some issues with the presentation of the qualitative conclusions in the article, I recommend that the paper be accepted after response to the below comments/improvements.
Major comments:
1. One of the regimes discussed is the “superradiant instability (SRI)”. It is unclear in the current presentation whether this process is indeed superradiant, or whether this is intended as descriptive terminology. There are more details in Sec. 4.2 discussing this point, but again the description appears mostly by analogy to the other studies in Refs. [5,13]. The authors have indeed identified the “possibility of having superradiant scattering”, but it unclear whether they can make the conclusion that the process is indeed superradiant.
2. In Figure 5, the rightmost plot extrapolates the behaviour of the large system to an infinite one. The description here is challenging to follow and it is not clear how the inferences to the infinite system are made. As a starting point, it would be good to produce an equivalent plot for the L_y=120 data. In particular, the origin of the blue line and its relation to the imaginary spectrum is unclear. Including the same for the L_y=120 data could make the connection more apparent. The discussion of the authors jumps between the large system and infinite system behaviour, and it cannot be sufficiently followed to understand the origins of the rightmost plot of Fig. 5, or how rigorously it follows from increasing the system size.
3. In Fig. 6, the authors investigate a system with absorbing boundaries, to simulate the effect of the infinite system where there are no edge reflections. On page 12, the authors mention that the density perturbations grow exponentially in time. This is not clear from any of the data presented and some additional plots or analysis is needed to support this statement. Again, in the discussion the “superradiant” aspects of this system are noted, but it is again unclear whether this extends beyond an analogy.
4. Page 13, “There is good agreement with the largest instability rates obtained for a finite but large system”. The agreement between the two data sets shown in the lower middle panel of Fig. 5 doesn’t appear particularly impressive – there is significant mismatch outside of error bars for the dominant instability.
Additional comments
5. Wording in the first sentence of the introduction “one of the most fundamental ones being” to “with one of the most fundamental being…”
6. The second paragraph of the introduction is difficult to understand as worded, perhaps defining a vortex sheet or better defining what is meant by “sharp” would help – clearly this is just that the shear layer size varies between the two regimes but would be helpful to state this.
7. In the third paragraph “see for example [3] for complete references”. It would be better to instead list relevant references here, rather than this nested approach which is not standard.
8. Second page “(RI) is analogous to the ergoregion instabilities…” without more being said this statement adds very little in terms of understanding. If the authors want to keep Ref. [14] here it requires a little more explanatory detail.
9. Page 4: There is a discussion about the shear layer width and its relation to the velocity, but there is no analysis of the shear layer in the GPE simulations. It would be useful to add a cross-section plots to Fig. 1, for the two velocities shown.
10. In the caption of Fig. 1 “clusterize” – not very standard terminology – just “cluster”?
11. End of page 4, there is a discussion on the shear configuration being a stationary state – yet the instability emerges in the GPE simulations shown. Some small discussion is needed on the seeding of the instability in the simulations – presume it is due to numerical noise?
12. Near Eqn. (8), page 6, \sigma_3 is not defined.
13. In the discussion following Eqns. (6-9), there is switching between \nu and \Delta\nu. It would seem that this is a typo, presumably the relative velocity \Delta\nu is the significant variable in all cases?
14. Fig. 3, \Gamma_{max} does not appear to be defined anywhere, similarly K_{max} is not explicitly defined.
15. Page 9, just after Eqn. (12), “red line in the rightmost lower plot” – the red line is in the leftmost plot.
We thank the referee for the careful reading of our manuscript and for the positive assessment of our results. In the following we reply point by point to their remarks, indicating the changes we made to the manuscript to address the issues they raised. We attach a pdf of the new manuscript with the main changes highlighted in red.
Major comments
-
The term superradiant was not intended as a descriptive terminology, the SRI stems from superradiant processes that occur in the system. We concluded this from the spectra, since the dynamical instabilities emerge from resonance of opposite-normed modes that are phononic propagating waves. However we probably have relied too much on our intuition in showing this. To back up our conclusions we added an Appendix (A1), with appropriate references in the main text, in which we report the result of numerical computations that show the occurrence of amplified scattering processes, for frequencies and Bloch momenta at which we found SRI. Our simulations involve the scattering of wavepackets in a system large enough that they do not interact with the boundaries. These simulations show that these sizable amplifications are stable processes in an infinite system, while they inevitably lead to SRI in a finite system.
-
We improved the discussion of the inference of the spectrum in the infinite system. In particular we included in the right plot of Figure 5 a plot of all the dynamically unstable modes of the large-but-finite system, that clearly distinguishes the instabilities that disappear for an unbound system from those that remain.
-
We changed Figure 6 to include a panel showing the exponential growth of (some of) the momentum components of the unstable growing perturbation. For what concerns the superradiant nature of the RI, we updated the discussion and, as for the SRI, we added in Appendix A2 a scattering numerical experiment in which we show how the RI is associated to superradiant scattering given by the existence of a negative-energy mode localized around the array of vortices. Differently from the SRI case, here the amplified scattering cannot occur without starting the associated instability. The mechanism is completely analogous to the one we illustrated for multiply quantized vortices in [20].
-
What we meant with good agreement is that the measured instability rates for the setup mimicking an infinite system are in correspondence and comparable to the ones obtained by diagonalizing a large but finite system. An exact match is actually not to be expected and the mismatch in the dominant instability bubble can be ascribed to a finite size effect. Comparing with the panel for $L_y=60\xi$ one can see that the dominant bubble is higher in that case. The dominant instability is hence still a decreasing function of the system size, and a further decrease in the $L_y\to\infty$ limit is to be expected. We improved the discussion about this point.
Additional comments 5-6. We improved the first two paragraphs 7. We listed directly the relevant references 8. We added more details about that comparison. The parallelism is further explained in Section 4.2, when dealing with the RI. 9. We added two panels to Figure 1, showing the velocity profile obtained numerically and a superimposed plot of the hyperbolic-tangent profile. 10. Me modified the term. 11. We added a footnote commenting on this. The way in which the shear layer is created via a time-dependent potential implies that fluctuations are present in the system. Also, as you pointed out, numerical noise is a further seed for instabilities. 12-15. We defined the undefined symbols, uniformed the notation to $\Delta v$ and corrected the wrong indication for the red line.

Anonymous on 2021-12-23 [id 2047]
Authors study the instability of a vortex street in a two-dimensional Bose-Einstein condensate (BEC) by numerically solving the Gross-Pitaevskii (GP) equation and the Bogoliubov equations. They initially prepare a Gaussian potential between two counter-propagating superflows and observe the nonequilibrium dynamics after removing the potential. The counter-propagating state decays into a vortex street and makes distinct dynamics depending on the initial relative velocity \Delta v between the two superflows; the non-equilibrium dynamics are dominated by complicated motion of vortices for the subsonic superflows while the acoustic excitations are radiated to the bulk for the supersonic case.
Authors focus the aspects of Kelvin-Helmholtz instability (KHI) and the superradiance due to dynamical instability of this system. I think the connection of the hydrodynamic instability with superradiance can be an interesting topic. However, the argument and discussion by authors do not have enough evidence which can prove their theory, especially, with respect to the Bogoliubov analysis. I would like to describe the reason why I think so below.
(1) Point vortex model
Authors try to analyze the stationary solution of an array of quantized vortices in terms of the theory of KHI in an unconventional formalism. This system should be considered more fundamentally as a vortex street (not Karman’ vortex street). It is also known that the system is unstable against a transverse shift of vortices (see some textbook on hydrodynamics). Then, I would like to ask authors to compare the accuracy of their numerical analysis with the result of the point vortex model by removing the trapping potential. I think the situation considered is too complicated to understand such a fundamental aspect of the system.
(2) Width of the shear layer
The smallest length scale, which determine hydrodynamics in a superfluid is the healing length \xi. However, the width \delta_v defined by Eq. (3) is no such a restriction. The interpretation of the width is crucial to characterize the fundamental property of this system because it is used in Eq. (11) for justifying the numerical analysis in Fig. 2.
A vortex street looks like a velocity shear layer if it is viewed on the length scale much longer than the distance between neighboring vortices. Therefore, I can agree with authors expectation that theory of KHI is applicable to this system under some condition. Concretely speaking, it is natural to expect that KHI would be realized when the wave number k_H of the unstable mode is much smaller than n_{vort}. However, the value of k_H, considered by authors, seems to be similar to n_{vort}~Mv/hbar as is mentioned in the last paragraph of Sec. 4.1.
Although I think this is the most fundamental aspect of this system to be examined first, authors start with unconventional formulation of KHI with no kind explanation. For example, the result of Eq. (10) looks quite different from the conventional result obtained by considering the stability of interface tension wave (I do not have the book of Ref [1]). However, there is no explanation about the difference in the manuscript. Furthermore, authors continues to extend the theory for the case with a finite width of the shear layer. The velocity field is not potential flow (vortex free) inside the shear layer in classical fluid, but it is fully potential flow in superfluids. How to connect the two systems? If authors try to apply approximately, what is the condition that can be safely applied?
(3) Interpretation of the results in Fig. 2
As authors suggest in the text, there could be the localized modes at the shear layer or vortices and the acoustic mode (phonons) propagating in the bulk fluid. Although a lot of branches are plotted for the real part in Fig. 2, there is no explanation about which branch corresponds to the localized or bulk modes. Thereby, the authors’ assertion that the resonance occurs between the bulk and localized modes lacks persuasiveness. This is contrast to the explanation on the resonance and the superradiance due to the dynamical instability of a multiply quantized vortex in superfluid in Ref. [14], or originally and more quantitatively by Takeuchi, et. al., J. Phys. Soc. Jpn. 87, 023601 (2018).
The style of the plot also makes it difficult for readers to understand. The difficulty might come from the introduction of Eq. (12) without enough explanation. Where does this relation come from? The relation of momentum and wave number can be complicated in the Bogoliubov analysis because of the complicated property of the norm in the presence of unstable mode with complex eigenvalue. This is related to the form of the Bovoliubov transformation and its normalization condition. The corresponding form of Eq. (6) is different from the conventional transformation, i.e., \delta\psi =u-v^*, which make readers confused more.
I am not also satisfied with the fact that the “adjustment” is not so small for the fitting curve in the leftmost panel of Fig. 2. In such a difficult situation, I am sorry that I could not read the content of Sec. 4.2 seriously.
(4) Previous works on KHI
Baggaley and Parker revealed the nonequilibrium dynamics for the case with small and large number of vortices in a very similar situation of single-component superfluids in Ref. [12]. Since authors investigate a very similar situation, they should explain distinction between the result of Ref. [12] and their result more clearly. Additionally, although authors give attention to the radiation of acoustic excitations propagating away in the bulk fluid for the supersonic case, a very similar phenomenon has been observed for KHI in immiscible binary BECs by Kokubo et. al. Phys. Rev. A 104, 023312 (2021). According to this paper, the acoustic excitations are shock waves forming Mach-cone-like structure. A similar pattern is also observed in the rightmost panel in Fig. 1. This is just a nonlinear effect although authors try to describe such acoustic radiation as superradiance in terms of the linear analysis based on the Bogoliubov theory. How to explain this inconsistency?

---

## Round 2 · Referee Report · Anonymous (Referee 2) · 2022-2-17

Report
Authors focus the aspects of Kelvin-Helmholtz instability (KHI) and the superradiance due to dynamical instability of this system. I think the connection of the hydrodynamic instability with superradiance can be an interesting topic. However, the argument and discussion by authors do not have enough evidence which can prove their theory, especially, with respect to the Bogoliubov analysis. I would like to describe the reason why I think so below.
(1) Point vortex model
Authors try to analyze the stationary solution of an array of quantized vortices in terms of the theory of KHI in an unconventional formalism. This system should be considered more fundamentally as a vortex street (not Karman’ vortex street). It is also known that the system is unstable against a transverse shift of vortices (see some textbook on hydrodynamics). Then, I would like to ask authors to compare the accuracy of their numerical analysis with the result of the point vortex model by removing the trapping potential. I think the situation considered is too complicated to understand such a fundamental aspect of the system.
(2) Width of the shear layer
The smallest length scale, which determine hydrodynamics in a superfluid is the healing length \xi. However, the width \delta_v defined by Eq. (3) is no such a restriction. The interpretation of the width is crucial to characterize the fundamental property of this system because it is used in Eq. (11) for justifying the numerical analysis in Fig. 2.
A vortex street looks like a velocity shear layer if it is viewed on the length scale much longer than the distance between neighboring vortices. Therefore, I can agree with authors expectation that theory of KHI is applicable to this system under some condition. Concretely speaking, it is natural to expect that KHI would be realized when the wave number k_H of the unstable mode is much smaller than n_{vort}. However, the value of k_H, considered by authors, seems to be similar to n_{vort}~Mv/hbar as is mentioned in the last paragraph of Sec. 4.1.
Although I think this is the most fundamental aspect of this system to be examined first, authors start with unconventional formulation of KHI with no kind explanation. For example, the result of Eq. (10) looks quite different from the conventional result obtained by considering the stability of interface tension wave (I do not have the book of Ref [1]). However, there is no explanation about the difference in the manuscript. Furthermore, authors continues to extend the theory for the case with a finite width of the shear layer. The velocity field is not potential flow (vortex free) inside the shear layer in classical fluid, but it is fully potential flow in superfluids. How to connect the two systems? If authors try to apply approximately, what is the condition that can be safely applied?
(3) Interpretation of the results in Fig. 2
As authors suggest in the text, there could be the localized modes at the shear layer or vortices and the acoustic mode (phonons) propagating in the bulk fluid. Although a lot of branches are plotted for the real part in Fig. 2, there is no explanation about which branch corresponds to the localized or bulk modes. Thereby, the authors’ assertion that the resonance occurs between the bulk and localized modes lacks persuasiveness. This is contrast to the explanation on the resonance and the superradiance due to the dynamical instability of a multiply quantized vortex in superfluid in Ref. [14], or originally and more quantitatively by Takeuchi, et. al., J. Phys. Soc. Jpn. 87, 023601 (2018).
The style of the plot also makes it difficult for readers to understand. The difficulty might come from the introduction of Eq. (12) without enough explanation. Where does this relation come from? The relation of momentum and wave number can be complicated in the Bogoliubov analysis because of the complicated property of the norm in the presence of unstable mode with complex eigenvalue. This is related to the form of the Bovoliubov transformation and its normalization condition. The corresponding form of Eq. (6) is different from the conventional transformation, i.e., \delta\psi =u-v^*, which make readers confused more.
I am not also satisfied with the fact that the “adjustment” is not so small for the fitting curve in the leftmost panel of Fig. 2. In such a difficult situation, I am sorry that I could not read the content of Sec. 4.2 seriously.
(4) Previous works on KHI
Baggaley and Parker revealed the nonequilibrium dynamics for the case with small and large number of vortices in a very similar situation of single-component superfluids in Ref. [12]. Since authors investigate a very similar situation, they should explain distinction between the result of Ref. [12] and their result more clearly. Additionally, although authors give attention to the radiation of acoustic excitations propagating away in the bulk fluid for the supersonic case, a very similar phenomenon has been observed for KHI in immiscible binary BECs by Kokubo et. al. Phys. Rev. A 104, 023312 (2021). According to this paper, the acoustic excitations are shock waves forming Mach-cone-like structure. A similar pattern is also observed in the rightmost panel in Fig. 1. This is just a nonlinear effect although authors try to describe such acoustic radiation as superradiance in terms of the linear analysis based on the Bogoliubov theory. How to explain this inconsistency?
- We thank the referee for their insight. In the following we respond to each of their remarks, that we think mostly stem from a misinterpretation of our aims when using results from hydrodynamics. We modified the text so to make the rationale of our comparisons more clear. We include here a pdf of the manuscript where the changes and additions are highlighted in red.
(1) Point vortex model Authors try to analyze the stationary solution of an array of quantized vortices in terms of the theory of KHI in an unconventional formalism. This system should be considered more fundamentally as a vortex street (not Karman’ vortex street). It is also known that the system is unstable against a transverse shift of vortices (see some textbook on hydrodynamics). Then, I would like to ask authors to compare the accuracy of their numerical analysis with the result of the point vortex model by removing the trapping potential. I think the situation considered is too complicated to understand such a fundamental aspect of the system.
We are sorry but we did not manage to find stability analyses of the vortex street you mention in the hydrodynamics textbooks we consulted. If you could give us some direct references we would be interested in looking at them. For the point of the present work however we feel that additional comparisons with hydrodynamic models are not needed. Our aim was to further characterize what was identified as KHI in [12] ([18] in the new manuscript), before dealing with the other instabilities we identified. We feel that the comparison with finite-width shear layers already shows a good parallelism with what is commonly called KHI. The main point we want to highlight is that the hydrodynamics results we report are not meant as being a good description of our system, it is just a reference model for a \textit{different system} that we take as a comparison for our numerical results. We tried to better highlight this in the text. We think it is interesting that many features of KHI are present also in our very different physical system.
(2) Width of the shear layer The smallest length scale, which determine hydrodynamics in a superfluid is the healing length $\xi$. However, the width $\delta_v$ defined by Eq. (3) is no such a restriction. The interpretation of the width is crucial to characterize the fundamental property of this system because it is used in Eq. (11) for justifying the numerical analysis in Fig. 2. A vortex street looks like a velocity shear layer if it is viewed on the length scale much longer than the distance between neighboring vortices. Therefore, I can agree with authors expectation that theory of KHI is applicable to this system under some condition. Concretely speaking, it is natural to expect that KHI would be realized when the wave number $k_H$ of the unstable mode is much smaller than $n_{vort}$. However, the value of $k_H$, considered by authors, seems to be similar to $n_{vort}\sim Mv/hbar$ as is mentioned in the last paragraph of Sec. 4.1.
Again we wish to stress that we are not using Eq. (11) (now eq. (12)) to \textit{justify} our numerical results, we are just taking the hydrodynamic model as a comparison to see which feature survive the very different setting. We agree that we consider values of the momenta of the perturbations comparable to the healing length, and that also the shear layer width is comparable to that. This in fact makes the hydrodynamic model not a good model for our system, that however we investigate directly by numerical means, and we take the hydrodynamics results as a comparison of what is usually understood as KHI.
Although I think this is the most fundamental aspect of this system to be examined first, authors start with unconventional formulation of KHI with no kind explanation. For example, the result of Eq. (10) looks quite different from the conventional result obtained by considering the stability of interface tension wave (I do not have the book of Ref [1]). However, there is no explanation about the difference in the manuscript. Furthermore, authors continues to extend the theory for the case with a finite width of the shear layer. The velocity field is not potential flow (vortex free) inside the shear layer in classical fluid, but it is fully potential flow in superfluids. How to connect the two systems? If authors try to apply approximately, what is the condition that can be safely applied?
We do not believe that the dispersion relation of the KHI we report is unconventional, since we also found it in other books. As an example we added a second book as a reference for the classical hydrodynamics results we mention (the one by Drazin and Reid [2]) where this result is also obtained in the introductory chapter. We added some specifications to give some missing information about the system for which that result is derived, that is an incompressible inviscid fluid of constant density without gravity. For what concerns the final questions we agree that the flow is not a potential one in the hydrodynamic model, while it surely is in the superfluid we are considering. But again, let us stress that we are not trying to apply that model to our system.
(3) Interpretation of the results in Fig. 2. As authors suggest in the text, there could be the localized modes at the shear layer or vortices and the acoustic mode (phonons) propagating in the bulk fluid. Although a lot of branches are plotted for the real part in Fig. 2, there is no explanation about which branch corresponds to the localized or bulk modes. Thereby, the authors’ assertion that the resonance occurs between the bulk and localized modes lacks persuasiveness. This is contrast to the explanation on the resonance and the superradiance due to the dynamical instability of a multiply quantized vortex in superfluid in Ref. [14], or originally and more quantitatively by Takeuchi, et. al., J. Phys. Soc. Jpn. 87, 023601 (2018).}
In Figure 2 the superradiant physics of the localized modes does not come significantly into play, since it matters for supersonic velocities and large systems, that we investigate in Section 4.2.2. In that context the presence of a localized mode is visible in Figure 6, and also in the newly added Appendix A2, where it can be also seen that negative-energy localized modes exist, very similarly to the vortex case (for which we added also the work you mentioned as a reference [15]). About the plots of Figure 2, we think it is not really possible to point at modes and say which is localized and which is not, since the output of the diagonalization are modes of the whole system, in which resonances between localized and phononic modes appear as a single mode.
The style of the plot also makes it difficult for readers to understand. The difficulty might come from the introduction of Eq. (12) without enough explanation. Where does this relation come from? The relation of momentum and wave number can be complicated in the Bogoliubov analysis because of the complicated property of the norm in the presence of unstable mode with complex eigenvalue. This is related to the form of the Bovoliubov transformation and its normalization condition. The corresponding form of Eq. (6) is different from the conventional transformation, i.e., $\delta\psi =u-v^*$, which make readers confused more.
The relation between hydrodynamic and Bloch momentum is a technical point depending on how one chooses to take the fluctuation field. This is what we explain in the discussion on page 9, that we tried to improve. This is not altered by the Bogoliubov dispersion, and in fact would be present also if we considered the so-called hydrodynamic approximation for fluctuations, that removes dispersion. It is a matter of convention and all the properties of the system, including dispersion and the presence of dynamical instabilities, come after and independently of this choice. Also, we do no think that our convention for the Bogoliubov transformation is unusual, this is the one used for example in the book by Pitaevskij and Stringari (ref. [24] section 5.6), that differs for a minus with respect to the one you wrote. This is also not related to out choice of the Bloch momentum. We added footnote 2 in page 9 to highlight these facts and try to avoid this confusion.
I am not also satisfied with the fact that the “adjustment” is not so small for the fitting curve in the leftmost panel of Fig. 2. In such a difficult situation, I am sorry that I could not read the content of Sec. 4.2 seriously.}
As we already pointed out, that curve is not a theoretical justification of our results, it just serves as a comparison with a model for a different system in which the KHI is usually studied. Moreover, the content of Section 4.2 does not mention those hydrodynamic models anymore, the analysis is purely based on our solutions of the Bogoliubov problem.
(4) Previous works on KHI Baggaley and Parker revealed the nonequilibrium dynamics for the case with small and large number of vortices in a very similar situation of single-component superfluids in Ref. [12]. Since authors investigate a very similar situation, they should explain distinction between the result of Ref. [12] and their result more clearly.}
Our work starts from the results of Baggaley and Parker. We first characterized in a different way the KHI they observed and we then proceeded to consider higher velocity differences, in which novel physical mechanisms emerge that, to our knowledge, were not reported before. In ref. [12] (now [18]) only velocity differences below $2 c_s$ were considered. Also, the drift of vortices at smaller velocities was not reported. We tried to make more clear the comparison with that work in the introduction.
Additionally, although authors give attention to the radiation of acoustic excitations propagating away in the bulk fluid for the supersonic case, a very similar phenomenon has been observed for KHI in immiscible binary BECs by Kokubo et. al. Phys. Rev. A 104, 023312 (2021). According to this paper, the acoustic excitations are shock waves forming Mach-cone-like structure. A similar pattern is also observed in the rightmost panel in Fig. 1. This is just a nonlinear effect although authors try to describe such acoustic radiation as superradiance in terms of the linear analysis based on the Bogoliubov theory. How to explain this inconsistency?
We think there is no inconsistency here. The Mach-cone-like structure of that work may well be something very similar to what we observe (we were not aware of that work, that we added to the references of previous works on the KHI). At least in our case however that is not a shock wave, since it emerges as small amplitude fluctuations. This is also confirmed by the fact that those features can be captured by the linear Bogoliubov theory. It is hence firstly a linear phenomenon, since it involves acoustic radiation, that only becomes nonlinear when the amplitude of the unstable modes become large enough.
Attachment:

---

## Round 3 · Referee Report · Anonymous (Referee 3) · 2022-2-21

Report

I cannot find a good reason why authors do not compare their numerical result with the analytic dispersion of the KHI. To ensure the reliability of numerical calculation by the comparison is crucial to judge the scientific quality of the current work. Since numerical results of the Bogoliubov equations with complex eigenvalues are normally quite complicated, it is difficult in general to give a proper physical interpretation to them, as in the current case. Additionally, high-precision calculation is required for diagonalizing the Bogoliubov equation because a small error can change the dispersion drastically in the case with complex eigenvalues. For example, the Nambu-Goldstone mode, which always exists in the presence of vortices and are localized at the vortex core, can be sensitive to a numerical error causing a unphysical complex mode.

The previous work by Baggaley et.al. is based on the Gross-Pitaevskii equations while the current work is on the Bogoliubov equations. In this situation, the comparison with the dispersion of the KHI is the fundamental step to do first for establishing the reliability before investigating more advanced, complicated situations. The simplest way to establish the reliability is to compare the numerical result with an analytic one if we have, and we have it; the numerical result can be compared with the dispersion of the KHI in fluid dynamics. Essentially, it does not matter which theory should be compared, the point vortex model or the model introduced by authors, if authors could make a proper introduction for it.

Finally, I am still confused with the formulation of the Bogoliubov transformation introduced by authors. The difference of the sign (plus or minus) in the front of the coefficient V_K is not essential. The transformation should be formulated with a matrix in Eq. (7). Or, does it include a matrix implicitly? This is the point I wanted to ask authors. I know the formulation, \dleta psi=U-(+)V^* and \delta psi^*=U^*-(+)V, but I am not familiar with the Bogoliubov "formulaiton" of the form, \dleta \psi =U_K and \delta\psi^*=V_K, introduced by authors.

  • validity: -
  • significance: -
  • originality: -
  • clarity: -
  • formatting: -
  • grammar: -

Author:  Luca Giacomelli  on 2022-03-30  [id 2341]

(in reply to Report 1 on 2022-02-21)
Category:
answer to question
reply to objection

We must say that we find it difficult to respond to the first referee. Their present objections on the comparison with the hydrodynamic results seem to us to be in contrast with the ones in their previous report.

They require a comparison of our numerical results with an analytical prediction, but such an analytical prediction is not available for our system. The analytical results from hydrodynamics we are reporting are not applicable to the present case, for the multiple reasons that also the first referee addressed in their first report (e.g. the shear layer in hydrodynamics is not irrotational). And, as we also explained in the response to the previous report, we are not using them as a good model of the physical situation under study.

We moreover feel that their concerns of the reliability of the results of our numerical diagonalizations are generic and do not point to a problem in particular. We agree that numerical diagonalizations in presence of complex-frequency modes can be delicate, however we performed many checks on our numerical results. As is good practice, for the diagonalizations as for the other codes, we checked the stability of our results with respect to variations of the numerical parameters (e.g. the spatial discretization and the integration step). Moreover we compared the results of the diagonalizations with the ones of time evolutions of the Bogoliubov problem, that are usually more robust. An example of such a comparison is visible in the $L_y=120\xi$ panel of Figure 5, where we compare the instability rates obtained via diagonalization with the ones extracted from a time evolution with a noisy initial condition. This shows that the complex-frequency modes we find are not spurious ones. A further check of consistency can be seen in the right panel Figure 5, where one can see that the real parts of the SRI unstable modes fall well inside the analytical red region for an infinite system.

Note that we had already done similar consistency checks with analogous codes for different physical setups in our previous publications, for instance in Ref.[21], where the instability rates for single vortices obtained with diagonalizations were confirmed with time-dependent simulations. This gives us full confidence on the reliability of our numerical calculations. To make this point fully clear to the reader, we added a footnote in Section 3 commenting on these consistency checks.

About the formulation of the Bogoliubov problem, we now understood what is the question of the referee and we thank them for pointing out this subtle point of a widely used approach. The formulation we are using is the one presented in detail in reference [27] (section 6.1.1), that we cite just before using it. By taking $\delta\Psi$ and $\delta\Psi^*$ as independent variables one recovers the linearity of the equations for the fluctuation field, that otherwise mix the field and its complex conjugate. This allows to obtain the spectrum via diagonalization of the Bogoliubov matrix of equation (9). This is a convenient mathematical treatment of the problem that however doubles the dimension of the space of the fluctuation field. In fact a mode $(U,V)$ with frequency $\omega$ has always a partner $(V^*,U^*)$ with frequency $-\omega^*$. After the diagonalization, to recover the physical fluctuation field one has to impose the conjugation relation between $\delta\Psi$ and $\delta\Psi^*$ (that were initially taken as independent). To this purpose it is enough to consider the sum of a pair of partner modes, so that $(\delta\Psi, \delta\Psi^*) = (U,V)+(V^*,U^*)=(U+V^*,V+U^*)$, which indeed satisfies the conjugation condition. This is the combination we take for example to obtain the density fluctuations shown for example in Figure 6. We added a footnote before equation (7) to explain this.

---

## Round 3 · Referee Report · Anonymous (Referee 4) · 2022-3-10

Report

The authors have improved the presentation of the paper and have clarified numerous points, while also making the presentation of the figures clearer. I am happy with the revised paper and recommend publishing.

Requested changes

As a minor comment, I found the added text in the caption of Fig. 1 made the following sentence, starting with “The first case shows the KHI behaviour…”, hard to follow. It is somewhat unclear what is referred to here by the authors in this sentence and the next. It would be clearer to refer to plots directly, i.e. “The upper plots shows the KHI behaviour…”.

---

## Round 3 · Author Response

Dear Editor,

We thank the referees for their insight and for helping us in making our work more complete and understandable. This new version of the manuscript includes several additions and changes to address the points raised by the referees.

We replied directly to the two reports and, given the impossibility of uploading a pdf with the resubmission, we included in our replies a pdf of the manuscript, in which the main changes are highlighted in red.

The main addition is an Appendix in which different kinds of superradiant scattering that occur in this system are demonstrated. This is intended to back up our conclusion of the superradiant nature of some of the instabilities we observe, whose discussion in the main text was also largely rewritten. Moreover, we updated some Figures to show more details and expanded the discussion of some key points, as requested by the first referee.

We also improved the exposition in several points, in particular those that we believe caused some misunderstanding with the second referee about the purpose of our comparisons with results from classical hydrodynamics.

We believe that these modifications, listed below, make our discussion clearer, display new interesting physics and reinforce our conclusions. We hence ask you to consider our paper for publication in SciPost Physics.

Best regards,

The authors

---

## Round 3 · List of Changes

A version of the manuscript in which these changes are highlighted in red was attached to the responses to the reports.

- We rephrased the parts of the introduction that were unclear
- We added two panels to Figure 1 showing the y dependence of the velocity, with a comparison with the hyperbolic tangent profile
- On page 5 we added a footnote to explain the seeding of the instabilities in the GPE computations
- We added a paragraph at the beginning of Section 4.1, better explaining the rationale of out comparison with the hydrodynamic results. We also added a second textbook reference for those results.
- We added some details to the discussion before Eq. (14) and a footnote explaining that the form in which we take the fluctuation field does not depend on the physical properties of our superfluid system.
- Because of its increased length, we separated Section 4.2 in two subsections
- In Section 4.2.1 we improved the discussion of SRI and added references to the Appendix, were additional computations displaying superradiant scattering of wavepackets are shown
- Section 4.2.2 was rewritten to address the points the referee raised about our inference on the infinite size system. Figure 5 was updated to show how this inference was made.
- Figure 6 was remade to include a panel showing the exponential growth of the unstable modes, from which the red circles shown in the Ly=120 panel Figure 5 are extracted. The mismatch between the diagonalization and the time-dependent calculations pointed out by one referee is now discussed.
-The superradiant nature of RI is now better discussed in Section 4.2.2, with references to the new Appendix A2.
-An appendix was added in which different kinds of superradiant scattering at the basis of SRI and RI are shown through additional numerical computations.

---

## Round 4 · Referee Report · Anonymous · 2022-4-14

Report
Dear Professor Davis,
I have carefully read the latest revision of the manuscript, before viewing the earlier reports. My overall impression of this manuscript is that it does open a new pathway in an existing research direction, with clear potential for multipronged follow-up work. I find the results on superradiant scattering of phonons from an array of vortices particularly interesting and worth publishing in SciPost Physics. In addition, the authors have provided detailed and satisfactory responses to all previous referee reports. Moreover, I found the added Appendices that clarify the physics of the superradiant scattering particularly welcome and helpful.
I suspect that the identity of the Referees 1 and 2 have been swapped between the first and second round of refereeing and I found it somewhat challenging to follow some points raised in the Anonymous Report 2 on 2022-2-17 and the Anonymous Report 1 on 2022-2-21. For instance, the request to compare the presented results with a point vortex model seems unjustified in this case where the most interesting new results are inherent to phonon radiation which cannot be modeled using point vortex models that assume complete absence of phonons. Many other points seemed to hinge on technicalities that do not affect the value of the presented key findings.
In summary, after reading the manuscript I would have recommended acceptance. After reading the referee reports and the authors' responses, I am not persuaded to change my original opinion and am therefore recommending to accept this manuscript for publication in SciPost Physics.
I would also like to share a few additional comments that are sufficiently minor that I can trust the authors' judgement on whether or not they require action or not.
1. perhaps the weakest point of this manuscript is how it is placed in the broader physics context as the referencing to previous works is quite scarce. For instance, it would be useful to link present work to the recent experimental observation of quantum Kelvin-Helmholz instability in a Bose gas
https://www.nature.com/articles/s41586-021-04170-2
2. in the first line of Chapter 2, the reference to a pancake shape made me incorrectly think of circular disk condensates. Perhaps the word pancake is not necessary here at all or at least it seems unnecessary to use italics to emphasize this word.
3. Looking at the equation (9) made me wonder if the -D operator on the lower diagonal of the matrix should be complex conjugated since it contains the -iK\partial_x operator?
4. In the context of Fig.2, the caption states that energetic instabilities begin to be present at the crossing of Dv = 2.0 c_s. However, there is a very large gap between Dv = 1.26 c_s and 2.42 c_s and already in the Dv = 1.26 c_s case there is one negative energy, positive norm, eigenstate.
5. on page 11, I think I am missing the point of the paragraph starting with "Notice...". The authors seem to highlight a difference between two systems based on the fact that they satisfy the same criterion Dv > 2 c_s?
6. in the context of [20,21], the instability of a doubly charged vortex was identified in https://doi.org/10.1103/PhysRevA.59.1533, experimentally observed in https://doi.org/10.1103/PhysRevLett.89.190403, and discussed as a resonance between positive and negative energy eigenstates for instance in
https://doi.org/10.1103/PhysRevA.68.023611
7. on page 16 in the context of [28], the mechanism of image vortices driving the bulk vortex motion seems to be equally well present in both uniform and harmonically trapped condensates with a boundary, as well as in the drifting case discussed in this work. In the harmonically trapped case the density gradient effect and the image vortex effect have been shown to be almost equally strong, https://doi.org/10.1103/PhysRevA.97.023617
Author: Luca Giacomelli on 2022-07-25 [id 2683]
(in reply to Report 1 on 2022-04-14)Here we attach the new version of the manuscript, with changes highlighted in red.
Attachment:
Author: Luca Giacomelli on 2022-07-22 [id 2677]
(in reply to Report 1 on 2022-04-14)
We thank the referee for their careful reading of our work and for their positive assessment of our results. In the following we reply to their comments, that we kept into account with some modifications of the manuscript.
-
We thank the referee for pointing out this recent work. We think that we connected our results with previous works on the KHI in superfluids in the introduction, where we cite the works we are aware of (i.e. [13-19]). We added reference [15] and the article the referee mentioned, commenting the connection with the rest of the literature.
-
The referee is right, the term can give rise to some confusion. We removed the word pancake, and we now refer only to a BEC tightly confined in one direction.
-
In equation (9) there is no complex conjugate on the lower diagonal element. This is due to the fact that this expression of the Bogoliubov problem was obtained by substituting equation (7) in the Bogoliubov problem featuring only real terms in the diagonal elements. The fact that $e^{iKx}$ is a global factor for the spinor implies that the term $-iK\partial_x$ will appear without conjugation in the lower diagonal elements.
-
The referee is right, strictly speaking the statement in our caption was incorrect. What we meant is that at the threshold $\Delta v=2c_s$ negative energy phononic waves begin to be present due to Landau instability. The negative energy states visible at the edge of the Brillouin zone are instead due to the dynamics of the single vortices and do not fall in the general picture we discussed associated to Fig. 2. They are instead responsible for the drift instability. Moreover, the threshold is not exactly $2c_s$ because the system is finite along $y$ and hence the phononic modes discrete. Thus the two parts of the spectrum begin to merge only at $\Delta v\simeq 2.42c_s$ and the spectra for $1.26c_s<\Delta v<2.42c_s$ are not particularly different from the first. We modified the caption and the discussion below Fig. 2 to make these points clearer.
-
With that comment we wanted to highlight a difference in the interpretation of negative-energy modes between the translationally invariant case one has in the absence of quantized vortices, and the present case. Reading that section again, we however noticed that this paragraph is not optimally placed and we have taken the opportunity of moving it to a footnote earlier in the section and slightly rephrasing it.
-
Regarding the instability of multiply charged vortices, we are aware that the literature on this topic is extremely large. However, it mostly deals with trapped condensates in which the role of the trap is unclear. On this basis we prefer to restrict our citations to works specifically dealing with vortices in unbound condensates that are most relevant for the physics we are discussing in our manuscript.
-
Thank you for pointing out that work, we were not aware of it. We changed our comment on the precession of vortices in trapped condensates and included that reference.
Author: Luca Giacomelli on 2022-07-25 [id 2682]
(in reply to Report 2 on 2022-04-16)We thank the referee for trying to close the gap between us. In the following we reply to the two main points they raised. We kept them into account with further minor modifications of the manuscript, to make our aims and our notation clearer. These changes are visible in red in the version of the manuscript we attach here. We however believe that these points stem from a mutual misunderstanding, are sufficiently minor ones and do not affect the main results on superradiant phenomena, that are presented in the rest of our work.
(i) Applicability of the hydrodynamic theory
What we did with the hydrodynamic prediction is not a fitting, rather a qualitative comparison. What we found interesting and wanted to show is that similar spectral features are shared by the two models, even though, as the referee correctly pointed out, the hydrodynamic model cannot take into account the presence of vortices and the single-particle nature of Bogoliubov excitations. As a side note the wavenumber $k_H\sim0.6/\delta$ is actually never reached since, as we argue in Section 4.1, the Brillouin zone is always smaller than this value.
The similarity of the spectral features in our opinion further justifies the name KHI for the leading instability we find in this regime, also considered the fact that similar instabilities have been called KHI in the literature simply from the nonlinear behavior of the system, without considering the spectral properties (e.g. in [19]).
We did not want to mislead readers into believing that numerical results are well described by the hydrodynamic theory. To make this clearer we modified the discussion below equation (13).
Anyway, we would like to stress once again that our discussion of the KHI is simply a further characterization at the spectral level of the instability that was first studied in [19] with the use of the Gross--Pitaevskii equation, and that the core of our new results are the superradiant instabilities we characterize in the following of the article, where no reference to the hydrodynamic model is made any more.
(ii) The form of the Bogoliubov transformation
First of all we would like to stress that the Bogoliubov problem we used is equivalent to the one the referee is referring to, as is discussed in [29]. All numerical data are obtained using equation (9), that seems to be familiar to the referee, so that our numerical diagonalization of the problem correctly diagonalizes the quadratic Hamiltonian and all the expected properties of the Bogoliubov modes hold.
We believe that the confusion may come from our notation in equation (7), that is strictly speaking not correct. The components of the spinor are $U_K$ and $V_K$ only after the diagonalization. To solve this confusion we changed equations (7) and (9) by introducing $\eta_K$ and $\chi_K$ to label the two components of the Bogoliubov spinor and reserved $U_K$ and $V_K$ to indicate the amplitudes of the eigenmodes, as indicated before equation (11) and in footnote 2 in Section 3, where the connection with the usual expression for the fluctuation field is made.
For what concerns the definition of momentum, the quantities $K$ and $k_H$ we introduce are actually not the momenta of the excitations but their wavenumbers. In particular $K$ is the Bloch wavenumber introduced in equation (7) and $k_H$ is the wavenumber of excitations along the shear layer in the hydrodynamic treatment. We changed the word \textit{momentum} to \textit{wavenumber} where we used it to refer to these quantities.
We would like to comment that we feel that this kind of approaches have been widely used in the literature and we think that this article is not the place to present these points more in detail, since references such as [29] are available.
Attachment:
redlined.pdf
Anonymous on 2022-08-02 [id 2705]
(in reply to Luca Giacomelli on 2022-07-25 [id 2682])The problem of (ii) has not been solved yet essentially. I can accept other revisions.
As mentioned, the interpretation of the dynamical instability as a resonance of positive- and negative-energy modes has been formulated by starting from the widely used formulation of \delta \psi =u+(-)v^* [more concretely for the current problem, \delta \psi =e^{iKx}(u+(-)v^*)]. I have not known literatures that formulate with the form of Eq. (7). Authors should show at least some references where such formulation is described clearly, although I think it is impossible.
I understand that this formulation is essential to connect the considered phenomenon with the superradiance. As I implied in the last report, the problem will be “resolved only by replacing the formula of Eq. (7) by the widely used form of the traditional Bogoliubov transformation.”
I found a typo in the added footnote on page 6;
\delta \psi=…+V^*e^{+iwt}
---> \delta \psi=…+V^*e^{+iw^*t}
This correction is also necessary for the formulation mentioned above.
Anonymous on 2022-09-02 [id 2786]
(in reply to Anonymous Comment on 2022-08-02 [id 2705])We thank the referee for pointing out the typo in the footnote, that we are going to correct. We are sorry that the referee feels we did not address their concerns. We however think that we did and that the discussion and the references we give in the article are sufficient to make this point clear.
As it is currently written, equation (7) is not a Bogoliubov transformation, but a definition of the spinor components. The trasformation is obtained with the diagonalization of equation (9): the resulting eigenvectors have as components the usual Bogoliubov amplitudes, that are connected to $\delta\psi$ with the usual Bogoliubov transformation written by the referee, as discussed in the footnote 2.
As we already pointed out, such an approach is explained in Section 6 of ref [29], where, after taking $\delta\psi$ and $\delta\psi^*$ as independent, the usual Bogoliubov transformation is obtained by expanding on the eigenmodes' spinors. See in particular equation (6.33) at page 69 of the journal version or equation (251) at page 72 of the arXiv version (arxiv.org/abs/cond-mat/0105058).
The Bogoliubov transformation we are performing is hence exactly the same the referee wrote and there is hence no difference in terms of spectral properties such as dynamical instabilities and positive and negative norm modes. Moreover, apart from the overall formulation, our calculations are performed with equation (9), that is very well known in the literature.

---

## Round 4 · Referee Report · Anonymous · 2022-4-16

Report
It seems that there is a gap between me and authors. To close the gap efficiently, I would like to point out only the two problems below, (i) and (ii), that have been already suggested in my first report. I will accept the revised manuscript for the publication if authors could answer to the problems clearly. In the previous report, I did make the criticism about the numerical computation but it is just the secondary problem. Thus, I will not mention any more about it for avoiding further unproductive communication.
(i) Authors should clarify the applicable condition of the hydrodynamic theory.
Authors make an artificial fitting of the numerical data with the hydrodynamic prediction of Eq. (13) in Fig. 2. There is no argument on how the way of the fitting is reasonable in the current situation. In my first report, I suggested that the fitting is not good. However, authors have made no reasonable answer for this problem. The fitting curve in Fig. 2 misleads readers into believing the numerical result are well-described by the hydrodynamic theory.
From the title of this manuscript, it is natural for reviewers to regard Kelvin-Helmholtz instability (KHI) as an important subject to be judged scientifically. At this stage, I have no judgement material without response from authors on the above problem. It is quite unclear how the wave number of k_H~0.6/\delta, as is mentioned just below Eq. (13), is reasonable for applying the hydrodynamic theory to the current system. Why is the theory reasonable despite a fact that the wavelength is comparable to the size of a quantized vortex? Additionally, a Bogoliubov excitation may have a single-particle behavior for such a large wave number because of the kinetic energy term in the Hamiltonian, where the hydrodynamic approximation can break down. The unclearness comes partly from the confusion of the Bogoliubov transformation and it will be discussed in the latter part of this report.
In this situation, we could not consider the hydrodynamic theory as “a reference model for a different system that we take as a comparison for our numerical results” as was stated by authors in the previous response. The unclearness might come from my misunderstanding of the hydrodynamic theory because I could not access the literatures [1,2]. I would be happy if authors could explain the point of the model more clearly and the connection with the current system quantitatively.
(ii) The form of the Bogoliubov transformation is confusing.
The Bogoliubov transformation and the Bogoliubov equation is originally introduced by Bogoliubov. The form of Eq. (9) is consistent with the original one but Eq. (5) with Eq. (7) is not. Since the consistency of Eq. (9) is the same as the original one, the numerical result in the manuscript could be accepted itself. On the other hand, the inconsistency from Eq. (7) can make an essential difference in giving the physical interpretation.
The Bogoliubov transformation is originally introduced to diagonalize the Hamiltonian (or the energy functional). The Bogoliubov equation is derived for this purpose and the problem is reduced to the eigenvalue problem. The linear combination of terms with exponents, e^{iwt} and e^{-iwt} with the eigenfrequency w, is crucial to achieve the diagonalization of the Hamiltonian; the form should be \delta\Psi= u+(-)v^*, where u \propto e^{iwt} and v \propto e^{iwt}. By inserting this formula into the original Hamiltonian, one obtains the contribution of the fluctuation to the energy as the form of Eq. (11).
The diagonalization of the Hamiltonian is essential to consider the fluctuation as elementary excitations with eigen energy. This idea is also crucial to this system because the dynamical instability is induced by a coupling between positive- and negative-energy excitations (modes). I wonder whether the fluctuation of Eq. (5) with Eq. (7) reproduces the contribution of Eq. (11) by inserting authors’ formula, \delta\Psi =U and \delta\Psi^* =U^*, into the energy functional or the Hamiltonian. I am sorry that I know only the widely used approach, which is historically used in the problem of dynamical instability in Bose-Einstein condensates, and I did not follow all the content of Ref. [27]. But, I am not sure the physical reason why \delta\psi and \delta\psi^* can be considered as independent for describing elementary excitations. They state just that “the functions \delta \psi and \delta \psi^* being now considered as independent” according to Ref. [27]. Even if they could succeed the diagonalization, I am not sure whether the formalism is consistent with the description of the perturbation theory for the dynamical instability, which derives a coupling between positive- and negative-energy modes as was revealed quantitatively in the problem of the splitting instability in Ref. [20]?
In addition to the formula of the excitation energy, I wonder the definition of the (pseudo-)momentum of excitation in their formalism. In the widely used approach, the momentum is defined as \hbar k N with the norm N=\int dxdy(|U|^2-|V|^2) and the wave number k of the excitation. This is computed as the contribution of the fluctuation to the total momentum of the condensate. This point could influence the interpretation of Eq. (14) and I want authors to clarify this problem too.
Finally, note that the problem of (ii), except for the problem of momentum just above, seems to be resolved only by replacing the formula of Eq. (7) by the widely used form of the traditional Bogoliubov transformation. This is because the mathematical treatment of U and V below Eq. (7) looks the same as the widely used one.

---

## Round 4 · Author Response

we resubmit the present article as you requested, with minor modifications to better explain the points raised by the First referee.
We kindly ask you to consider this new version of the manuscript for publication in SciPost Physics.
Sincerely,
The authors

---

## Round 4 · List of Changes

- We modified the caption of Figure 1 to make more clear which panels we are referring to, as suggested by the Second referee
-We added a footnote before equation (7) to explain the formulation of the Bogoliubov problem we are using
- We added another footnote in Section 3 to comment on the numerical checks we performed

---

## Round 5 · Author Response

Dear Editor,

thank you for handling our submission and for giving our work the possibility of further consideration from a third referee.

We thank the referees for the careful reading of our work. We responded directly to their reports and took into account their comments by making minor modifications to the manuscript. We added some references that were pointed out and added some comments to clarify the points that were raised. We attached to our responses the new manuscript with the changes highlighted in red.

For what concerns the first point raised in Report 2 we tried our best to get to a common point with the referee whose point of view is however quite different from ours: on one hand, they would like a detailed study of the applicability of the hydrodynamic theory; on the other hand, our aim is not to apply the hydrodynamic theory but to directly use the microscopic theory that best describes our system. We also think that the main new results in our work are in the following of the article, where we do not mention the hydrodynamic theory any longer. We hence think that a detailed study of the regime in which the hydrodynamic theory holds would fall outside the scope of the present article. As such, this problem is best addressed in a future independent work.

We think that with these new additions our article is even clearer and we hope that it can now be accepted for publication in SciPost Physics.

Best regards,

The authors

---

## Round 5 · List of Changes

• In the introduction we added a comment of the work mentioned by the first referee in their first point
  • In section 3 we changed the notation in equation (7) and in the following equations to address the potential confusion that the second referee pointed out
  • In the caption of Figure 2 and in the following discussion we corrected the previous statement that said that negative energy modes start to be present at $\Delta v=2c_s$. We added more details on the fact that the general picture we give there does not capture the full physics of the system, that is fully explored in the following sections.
  • In Section 4.1 we added a couple of sentences to make clearer that we are not applying the hydrodynamic theory to our system and that we are not using it to benchmark our results.
  • In Section 4.2.1, following the comment of the first referee, we moved a paragraph that was not well placed to footnote 5 earlier in the text.
  • in Section 4.3 we updated our comment on the precession of vortices in trapped condensates, including the new reference suggested by the first referee.

---

## Editorial Decision

published